# Ligand-tethered lipid nanoparticles for targeted RNA delivery to treat liver fibrosis

Xuexiang Han [1,8], Ningqiang Gong [1,8], Lulu Xue [1], Margaret M. Billingsley[1], Rakan El-Mayta[1], Sarah J. Shepherd[1], Mohamad-Gabriel Alameh [2,3], Drew Weissman[2,3] & Michael J. Mitchell [1,3,4,5,6,7] ✉

Lipid nanoparticle-mediated RNA delivery holds great potential to treat various liver diseases. However, targeted delivery of RNA therapeutics to activated liver-resident fibroblasts for liver fibrosis treatment remains challenging. Here, we develop a combinatorial library of anisamide ligand-tethered lipidoids (AA-lipidoids) using a one-pot, two-step modular synthetic method and adopt a two-round screening strategy to identify AA-lipidoids with both high potency and selectivity to deliver RNA payloads to activated fibroblasts. The lead AA-lipidoid AA-T3A-C12 mediates greater RNA delivery and transfection of activated fibroblasts than its analog without anisamide and the FDA-approved MC3 ionizable lipid. In a preclinical model of liver fibrosis, AA-T3A-C12 enables ~65% silencing of heat shock protein 47, a therapeutic target primarily expressed by activated fibroblasts, which is 2-fold more potent than MC3, leading to significantly reduced collagen deposition and liver fibrosis. These results demonstrate the potential of AA-lipidoids for targeted RNA delivery to activated fibroblasts. Furthermore, these synthetic methods and screening strategies open a new avenue to develop and discover potent lipidoids with targeting properties, which can potentially enable RNA delivery to a range of cell and tissue types that are challenging to access using traditional lipid nanoparticle formulations.

Liver fibrosis is an abnormal wound-healing response to chronic liver injury that affects millions of people worldwide and can further progress to liver cirrhosis and hepatocellular carcinoma[1–4], yet there are currently no approved antifibrotic therapies. Liver fibrosis is characterized by excessive accumulation of collagen-rich extracellular matrix (ECM). Hepatic stellate cells (HSCs), a liver-resident fibroblast population located in the space between liver sinusoidal endothelial cells (LSECs) and hepatocytes, are the main effector cells during liver fibrogenesis[3]. Upon liver injury, quiescent HSCs become activated and

trans-differentiated into proliferative, profibrogenic, and contractile myofibroblasts that secrete excessive ECM components, especially collagen[5,6]. Heat shock protein 47 (HSP47) is a collagen-specific molecular chaperone that plays a crucial role in the proper folding, assembly, and secretion of collagen into the extracellular space[7]. Hepatic expression of HSP47 is markedly increased in the fibrotic liver, and activated HSCs are identified as the primary source of HSP47[8], indicating that overexpression of HSP47 in activated HSCs accelerates the progress of this disease by supporting collagen biogenesis[9]. Thus,

[1]Department of Bioengineering, University of Pennsylvania, Philadelphia, PA 19104, USA. [2]Department of Medicine, University of Pennsylvania, Philadelphia, PA 19104, USA. [3]Penn Institute for RNA Innovation, Perelman School of Medicine, University of Pennsylvania, Philadelphia, PA 19104, USA. [4]Abramson Cancer Center, Perelman School of Medicine, University of Pennsylvania, Philadelphia, PA 19104, USA. [5]Institute for Immunology, Perelman School of Medicine, University of Pennsylvania, Philadelphia, PA 19104, USA. [6]Cardiovascular Institute, Perelman School of Medicine, University of Pennsylvania, Philadelphia, PA 19104, USA. [7]Institute for Regenerative Medicine, Perelman School of Medicine, University of Pennsylvania, Philadelphia, PA 19104, USA. [8]These authors contributed equally: Xuexiang Han, Ningqiang Gong. ✉e-mail: mjmitch@seas.upenn.edu

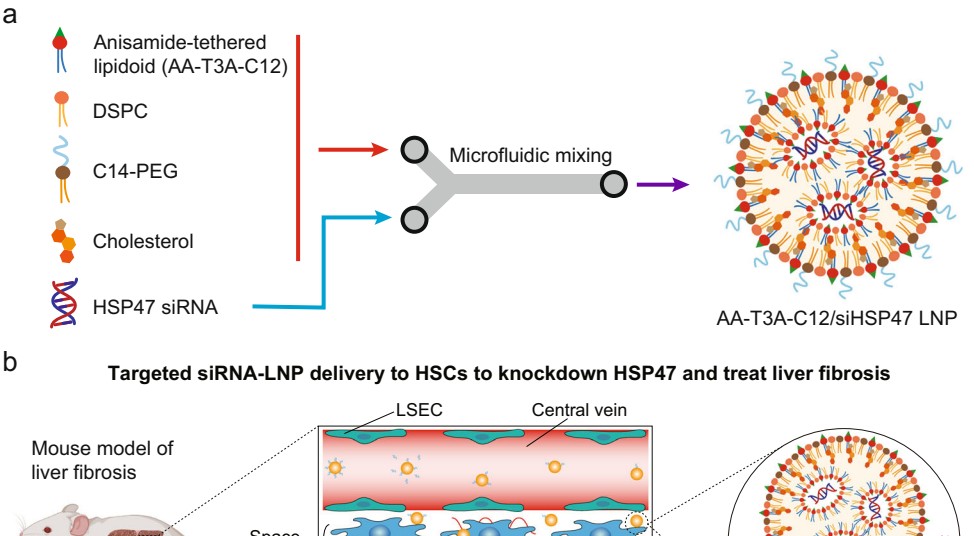

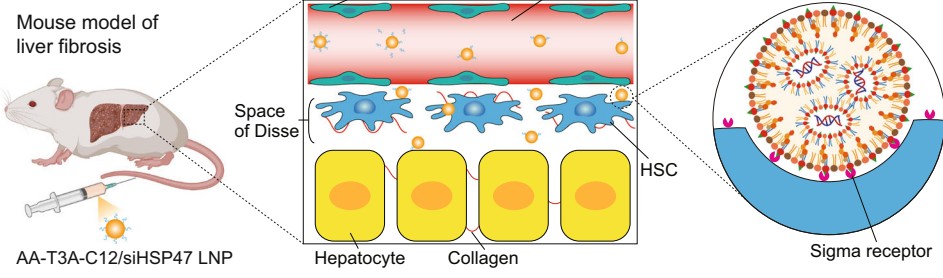

**Fig. 1 | Preparation and application of ligand-tethered lipidoid nanoparticles for targeted siRNA delivery to HSCs to treat liver fibrosis. a** Formulation of AA-T3A-C12/siHSP47 LNP via microfluidic mixing. The ethanol lipid solution containing anisamide-tethered lipidoid (AA-T3A-C12), phospholipid (DSPC), PEG-lipid (C14-PEG), and cholesterol is rapidly mixed with an acidic aqueous solution containing HSP47 siRNA in a microfluidic device to formulate AA-T3A-C12/siHSP47 LNP. **b** Scheme of targeted AA-T3A-C12/siHSP47 LNP delivery to activated HSCs to knockdown HSP47 and treat liver fibrosis. HSCs are located in the space of Disse, an area between LSECs and hepatocytes. After rapidly shedding PEG in circulation, the LNP exposes multivalent anisamide ligands on its surface that can strongly bind with sigma receptors overexpressed on activated HSCs to mediate cellular uptake. **b** was created with BioRender.com.

HSP47 represents a promising target for anti-fibrotic therapy[10]. With no specific HSP47 inhibitors available[11], utilizing small interfering RNA (siRNA) to silence HSP47 represents an attractive therapeutic strategy to reduce collagen production and alleviate fibrosis[9,12]. However, targeted and potent delivery of siRNA to activated HSCs remains challenging.

siRNA therapeutics face many extracellular and intracellular barriers[13,14], which necessitates the need for a delivery platform to achieve their potent delivery. Lipid nanoparticles (LNPs) are the most clinically advanced non-viral nucleic acid delivery platform, with the successful translation of one siRNA therapeutic and two mRNA vaccines[13,15]. LNPs are multi-component systems that are typically comprised of ionizable lipid (or lipidoid), cholesterol, phospholipid, and polyethylene glycol (PEG)–lipid (Fig. 1a)[16]. The cholesterol and phospholipid components contribute to the stability of LNPs and facilitate membrane fusion[17,18]. The PEG-lipid with short acyl chains stabilizes LNPs during formulation and storage but detaches from LNPs rapidly to promote cellular uptake upon intravenous injection[19,20]. The lipidoid is a key component to protect RNA therapeutics and drive endosomal escape for successful cytosolic delivery[15,21]. In addition, the lipidoid greatly influences LNP tropism and transfected cell types in vivo[22,23]. For example, by using different lipidoids, LNPs have successfully delivered RNA therapeutics to several types of liver cells, including hepatocytes, LSECs, and Kupffer cells[24–26]. To the best of our knowledge, lipidoids that intrinsically mediate targeted RNA delivery to activated HSCs have not been reported, although some LNPs have been demonstrated to passively target these cells[27,28].

We hypothesized that incorporating a small molecule ligand with high affinity for activated HSCs into the lipidoid molecule itself could enable targeted siRNA-LNP delivery to these cells, as the rapid shedding of PEG in circulation will expose multivalent ligands on the surface of LNPs that can strongly bind with overexpressed receptors and mediate cellular uptake (Fig. 1b)[20,23,29]. Among various ligands that have been successfully used for HSC-targeted drug delivery[30,31], we selected the neutral and stable anisamide as the building block to construct anisamide-tethered lipidoids (AA-lipidoids). Notably, anisamide is a high-affinity ligand for the sigma receptor that is highly expressed on rapidly proliferating activated fibroblasts, including activated HSCs[32–34]. In this study, we first developed a one-pot, two-step synthetic method to enable parallel synthesis of a combinatorial library of AA-lipidoids. After two rounds of screening, we identified the top-performing AA-lipidoid, AA-T3A-C12, with both high potency and selectivity for activated fibroblast transfection. In a mouse model of carbon tetrachloride (CCl$_4$)-induced liver fibrosis, HSP47 siRNA (siHSP47)-loaded AA-T3A-C12 LNP achieved ~65% knockdown and dramatically reduced liver fibrosis, which significantly outperformed the benchmark DLin-MC3-DMA (MC3) LNP.

## Results

### Design and synthesis of AA-lipidoids

In order to incorporate anisamide into lipidoids and enable parallel synthesis of a series of AA-lipidoids, a one-pot, two-step synthetic method was developed through the combination of the amine–succinimide coupling reaction and the ring-opening reaction between the epoxide and the amine, due to the simplicity and compatibility of these two reactions[35]. Briefly, the targeting ligand precursor anisoyl-N-hydroxysuccinimide (anisoyl-NHS, Figs. S1 and S2) was coupled to a polyamine core via an amide bond, whose free amines were subsequently substituted by epoxide tails (Fig. 2a). Such a tandem and modular synthetic strategy dramatically simplify synthesis, and the resultant materials can be used directly for in vitro screening without purification.

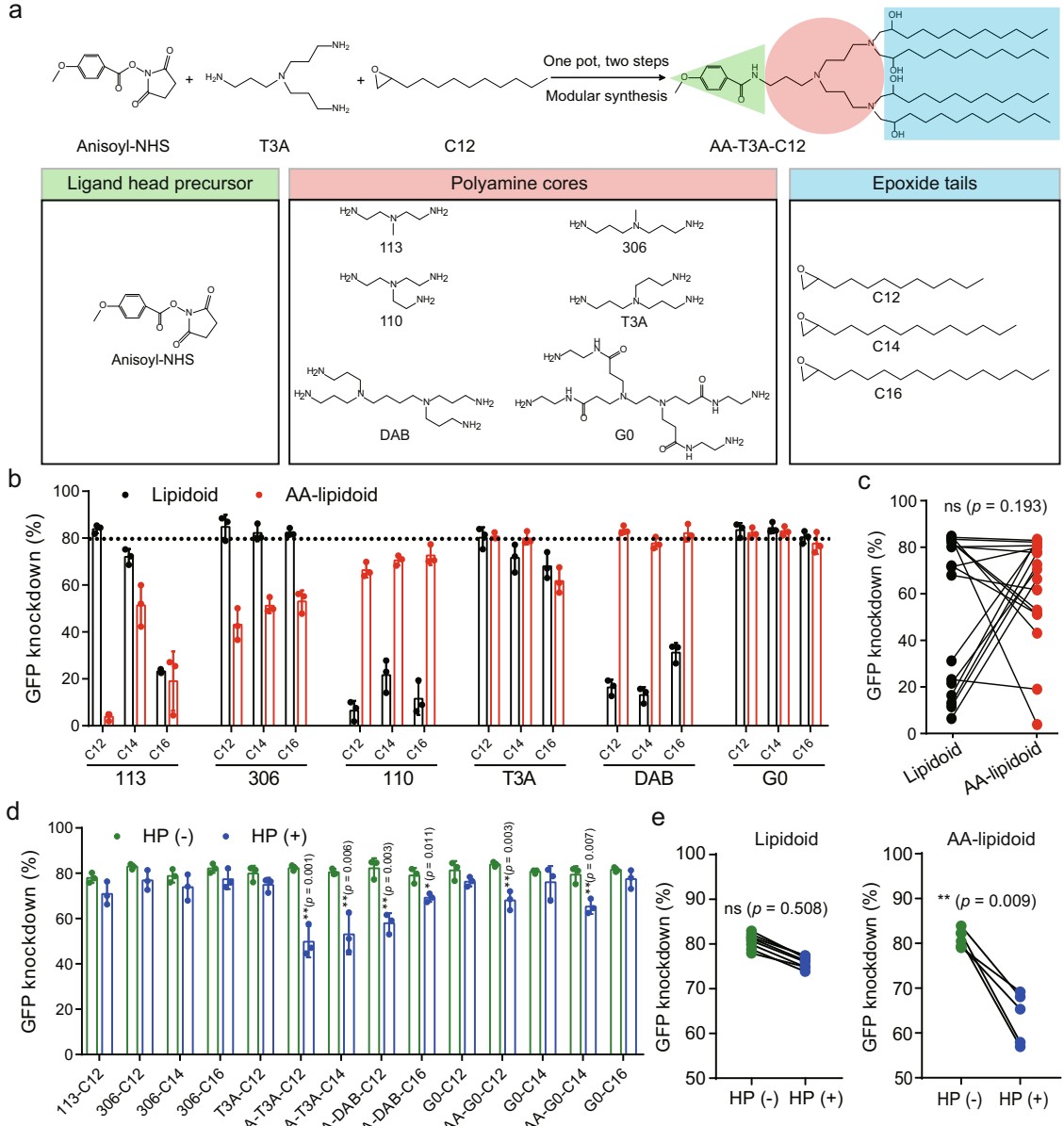

**Fig. 2 | Synthesis and screening of AA-lipidoids for targeted RNA delivery to activated fibroblasts. a** One-pot, two-step modular synthesis of AA-lipidoids. A representative synthesis of AA-T3A-C12 is shown. Anisoyl-NHS, polyamines, and epoxides were used to build a combinatorial library of 18 AA-lipidoids. **b** First-round screening of lipidoids and AA-lipidoids with high potency (*n* = 3/group). Lipidoids without anisamide were synthesized by the traditional ring-opening reactions between epoxides and polyamines. GFP siRNA-loaded LNPs were formulated to treat activated 3T3-GFP fibroblasts for 48 h to obtain their knockdown efficiency. The dashed line indicates 80% GFP knockdown. **c** Statistical analysis of structure–activity relationships. GFP knockdown efficiency was plotted based on lipidoids with or without anisamide. **d** Second round screening of lipidoids and AA-lipidoids with high dependency on sigma receptor-mediated transfection (*n* = 3/ group). Activated 3T3-GFP fibroblasts were pre-treated with haloperidol (HP) to block sigma receptors before treatment with GFP siRNA-loaded LNPs. **e** Statistical analysis of the relationship between sigma receptor blocking and knockdown efficiency. GFP knockdown efficiency was plotted based on treatment with or without HP. Data are presented as mean ± SD. ns not significant; *p < 0.05, **p < 0.01. **c–e** two-sided *t*-test. Source data are provided as a Source Data file.

## In vitro screening of AA-lipidoids

Initially, three representative polyamine (piperazine derivative 200, linear amine 114, and branched amine 110)-derived lipidoids with or without anisamide incorporation were evaluated for in vitro GFP silencing to assess which types of polyamines warranted further investigation (Fig. S3). Activated 3T3-GFP fibroblasts were obtained by pro-fibrotic transforming growth factor-β (TGF-β) stimulation, which enhances sigma receptor expression[33]. Pilot screening of epoxide C12-tailed lipidoids and AA-lipidoids showed that the branched amine 110-derived AA-110-C12 mediated ~70% GFP knockdown in activated 3T3-GFP fibroblasts, outperforming its counterpart lipidoid 110-C12 without anisamide as well as two benchmarks, 200-C12 (also known as C12-200)[35] and the commercially available DharmaFECT transfection reagent.

Therefore, additionally branched polyamines and epoxide tails were incorporated to expand the AA-lipidoid library (Fig. 2a). In total, 18 AA-lipidoids and 18 counterpart lipidoids were synthesized, formulated into 36 LNPs along with excipients (cholesterol, DSPE, C14-PEG) and GFP siRNA (siGFP), and subjected to high-throughput screening in activated 3T3-GFP fibroblasts. No significant cytotoxicity was observed for all LNPs after 48 h treatment (Fig. S4). Interestingly, less branched polyamine 113 and 306-derived AA-lipidoids showed

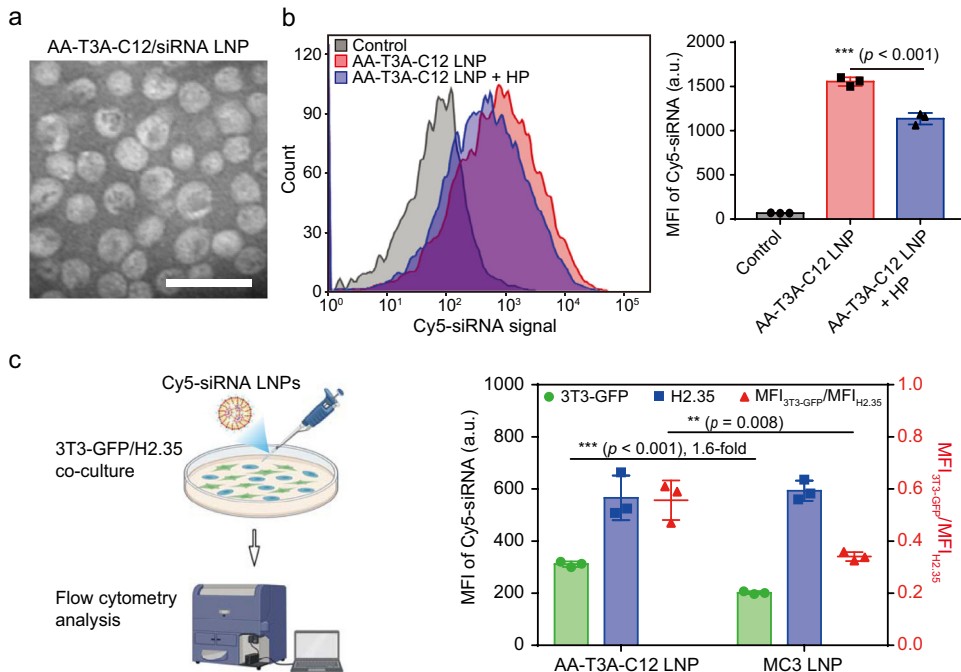

**Fig. 3 | Characterization of AA-T3A-C12 LNP and cellular uptake. a** A representative TEM image of AA-T3A-C12/siRNA LNP from three independent experiments. Scale bar, 100 nm. **b** Flow cytometry analysis of cellular uptake of Cy5-siRNA-loaded LNPs with or without haloperidol (HP) pretreatment (representative dataset from $n = 3$/group). **c** Flow cytometry analysis of competitive cellular uptake of Cy5-siRNA-loaded LNPs in a fibroblast/hepatocyte (3T3-GFP/H2.35) co-culture

environment (representative dataset from $n = 3$/group). The mean fluorescence intensity ratio between fibroblast and hepatocyte (MFI$_{3T3-GFP}$/MFI$_{H2.35}$) was calculated to indicate preferential uptake by fibroblasts over hepatocytes. Data are presented as mean ± SD. **$p < 0.01$; ***$p < 0.001$. **b** and **c** one-way ANOVA with Tukey's correction. Source data are provided as a Source Data file. **c** was created with BioRender.com.

reduced knockdown efficiency compared to their counterpart lipidoids regardless of epoxide tails, while more branched polyamine 110, T3A, DAB, and G0-derived AA-lipidoids either maintained or dramatically enhanced knockdown efficiency (Fig. 2b). One possible reason is that since the 113 and 306 cores are comprised of fewer amines, consumption of a primary amine by anisamide conjugation could affect their structure and ionization ability more profoundly than other cores with more amines. Nevertheless, an analysis of structure–activity relationships demonstrated that the incorporation of anisamide overall did not compromise the potency of lipidoids (Fig. 2c). In this round of screening, 6 AA-lipidoids and 8 lipidoids were identified to be highly potent with GFP knockdown efficiency above 80%.

These top 14 lipidoid candidates achieving >80% GFP knockdown were then subjected to a second-round screening to evaluate their dependency on sigma receptor-mediated transfection. Haloperidol (HP), a sigma receptor antagonist[36], was used to treat activated 3T3-GFP fibroblasts before LNPs treatment. For lipidoids without anisamide, no obvious loss of silencing activity was observed after HP treatment ($p = 0.508$, Fig. 2d, e). Similar results were observed when the FDA-approved MC3 LNP formulation was tested (Table S1 and Fig. S5). However, all AA-lipidoids showed significantly decreased knockdown efficiency after HP treatment ($p = 0.009$, Fig. 2d, e). Among the 6 AA-lipidoids, AA-T3A-C12 demonstrated the most significant loss of activity after sigma receptor blocking, which was chosen for subsequent studies due to its high dependency on sigma receptor-mediated transfection. These results suggest that after a first round of screening for efficiency and a second round of screening for selectivity, a potent AA-lipidoid (AA-T3A-C12) with active targeting ability was identified for further investigation.

## Characterization of AA-T3A-C12 and LNPs

The modularly synthesized lead lipidoid AA-T3A-C12 contains one anisamide head and four epoxide C12 tails, which are bound by the T3A

core (Fig. 2a). The structure of purified AA-T3A-C12 was confirmed by mass spectrometry and proton nuclear magnetic resonance (Fig. S6 and S7). Afterward, the four-component AA-T3A-C12/siRNA LNP was formulated by microfluidic mixing at an ionizable lipid:siRNA weight ratio of 10:1[37–39]. This weight ratio was chosen based on the results of both the gel retardation assay and the RiboGreen RNA assay, which resulted in high siRNA encapsulation efficiency (EE = 87.4 ± 3.8%, Fig. S8 and Table S1). The hydrodynamic diameter of these LNPs was ~65.6 nm with a narrow polydispersity index (PDI) of 0.018, which was slightly larger than empty LNPs (Table S1). Moreover, these LNPs had a neutral surface charge and a p$K_a$ of 5.72. Transmission electron microscopy (TEM) images showed that both empty LNPs and siRNA-loaded LNPs had a spherical morphology (Figs. 3a and S9), with some collapsed LNPs due to the dehydration process of sample preparation. Additionally, these LNPs had high colloidal stability, as large LNP aggregates (>200 nm) were not observed in the protein- or serum-supplemented environment at 37 °C for 48 h (Fig. S10). Overall, the above results suggest the formation of homogeneous and stable LNPs with high siRNA EE that are promising for potent delivery in vitro and in vivo due to their small size (<100 nm), neutral charge, and suitable p$K_a$ (5.5–7.0)[40].

## Targeted RNA delivery to activated fibroblasts using AA-T3A-C12 LNP

We then analyzed the cellular uptake of AA-T3A-C12 LNPs in activated fibroblasts using Cy5-siRNA as a cargo. Flow cytometry analysis showed that the cellular uptake of AA-T3A-C12 LNP was dramatically reduced after HP treatment ($p < 0.001$, Fig. 3b), which explained its reduced gene knockdown efficacy after sigma receptor blockade in Fig. 2d. Moreover, the cellular uptake of AA-T3A-C12 LNP was greater in TGF-β-stimulated 3T3 fibroblasts with overexpressed sigma receptors than in non-stimulated 3T3 fibroblasts ($p < 0.001$, Fig. S11), further confirming the enhanced uptake of AA-T3A-C12 LNP by activated fibroblasts.

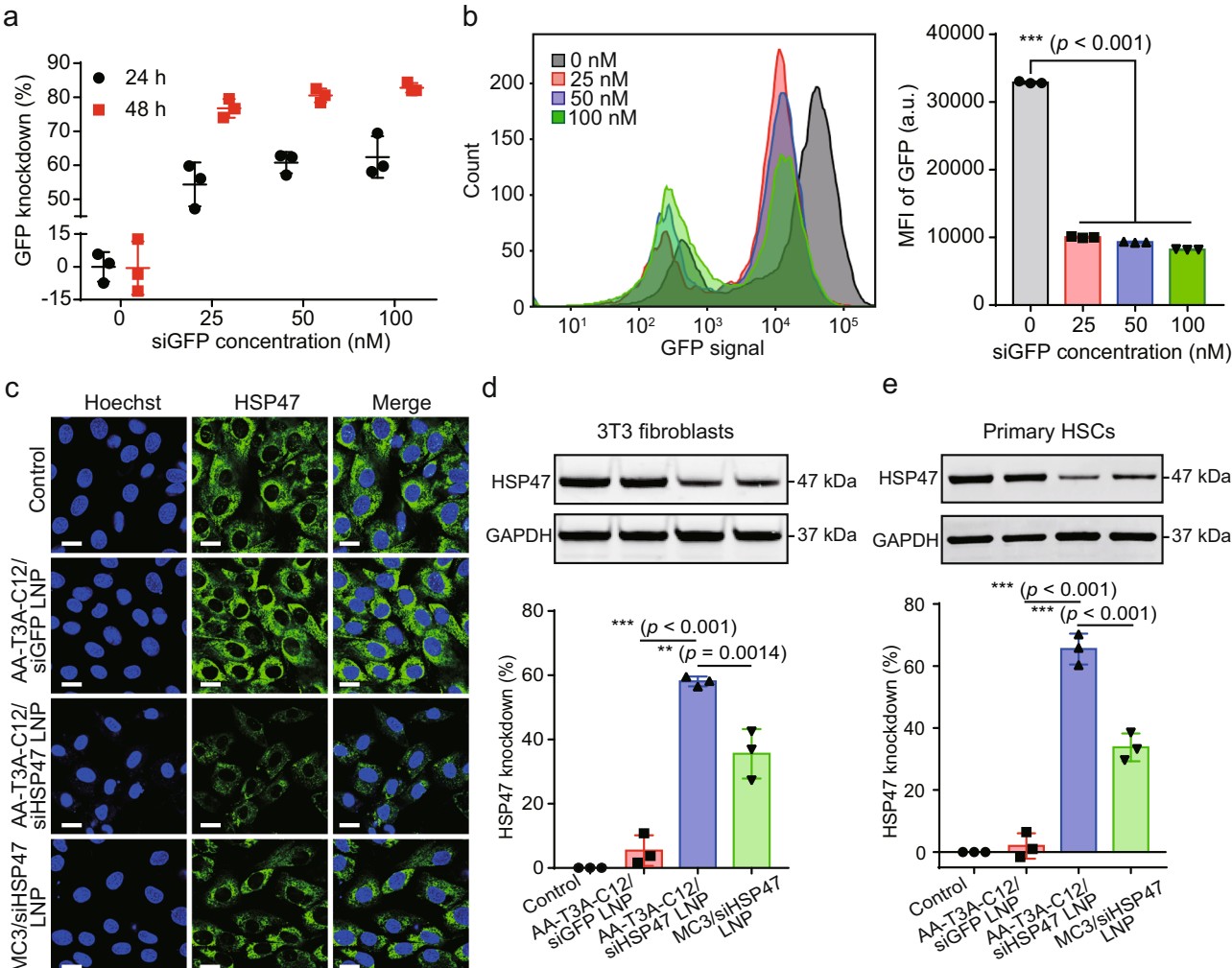

**Fig. 4 | AA-T3A-C12 LNP-mediated GFP and HSP47 knockdown in activated fibroblasts. a** GFP knockdown using AA-T3A-C12/siGFP LNP (*n* = 3/group). Activated 3T3-GFP fibroblasts were treated with AA-T3A-C12/siGFP LNP at the indicated dose for 24 or 48 h. **b** Flow cytometry analysis of GFP expression after AA-T3A-C12/siGFP LNP treatment for 48 h (representative dataset from *n* = 3/group). **c** Immunofluorescence (IF) staining of HSP47 in LNP-treated activated 3T3 fibroblasts. Scale bar: 20 μm. **d** and **e** Western blot analysis of HSP47 expression in LNP-treated activated 3T3 fibroblasts and primary HSCs (representative dataset from *n* = 3/group). GAPDH was used as an internal control. Quantitative analysis was performed using ImageJ software. Data are presented as mean ± SD (*n* = 3). **p* < 0.01; ****p* < 0.001. **b, d, e** one-way ANOVA with Tukey's correction. Source data are provided as a Source Data file.

Targeted delivery of AA-T3A-C12 LNP to activated fibroblasts was further investigated in a fibroblast/hepatocyte (3T3-GFP/H2.35) co-culture environment (Figs. 3c and S12), which intends to mimic the competitive cellular uptake of LNPs in the fibrotic liver by these two cell populations. MC3 LNP was included as a benchmark control. Flow cytometry analysis of cellular uptake showed that AA-T3A-C12 LNP achieved 1.6-fold greater Cy5-siRNA delivery to activated fibroblasts compared to MC3 LNP (*p* < 0.001), but both LNPs achieved similar delivery to hepatocytes (Fig. 3c). Additionally, AA-T3A-C12 LNP outperformed its non-targeted counterpart T3A-C12 LNP in delivering Cy5-siRNA to activated fibroblasts but not hepatocytes (Fig. S13). The calculated mean fluorescence intensity ratio between fibroblast and hepatocyte (MFI$_{3T3\text{-}GFP}$/MFI$_{H2.35}$, an indicator of fibroblast selectivity) was 0.34 for MC3 LNP, but a much higher ratio of 0.56 was observed for AA-T3A-C12 LNP (*p* = 0.008, Fig. 3c). Since fibroblasts are difficult to transfect and reluctant to engulf foreign substances in comparison to hepatocytes, a ratio of MFI$_{3T3\text{-}GFP}$/MFI$_{H2.35}$ below 1 is expected. Nevertheless, the above results suggest that AA-T3A-C12 LNP has better fibroblast selectivity and mediates greater siRNA delivery to activated fibroblasts than both the benchmark MC3 LNP and non-targeted T3A-C12 LNP.

Moreover, AA-T3A-C12 LNP-mediated enhanced transfection of activated fibroblasts was further confirmed by luciferase mRNA delivery (Fig. S14), as AA-T3A-C12 LNP achieved 2.6-fold higher luciferase expression in fibroblasts compared to MC3 LNP, but 1.4-fold lower expression in hepatocytes. This result indicates AA-T3A-C12 LNP is potentially promising for targeted delivery of large genetic constructs to activated fibroblasts as well.

## AA-T3A-C12 LNP-mediated robust gene knockdown in activated fibroblasts

After confirming AA-T3A-C12 LNP-mediated targeted gene delivery to activated fibroblasts, its potency to silence the gene was then investigated. In activated 3T3-GFP fibroblasts, AA-T3A-C12/siGFP LNP achieved time- and dose-dependent GFP knockdown in a highly efficient manner (Fig. 4a). After treatment with 50 nM of AA-T3A-C12/siGFP LNP for 48 h, >80% GFP silencing was achieved without noticeable cytotoxicity (Fig. S15), which is in agreement with the high-throughput screening results (Figs. 2b and S4). The GFP silencing effect of this LNP was further confirmed by flow cytometry analysis of GFP expression, with potent GFP knockdown achieved at a siGFP dose as low as 25 nM (Fig. 4b).

We next explored the potential of siHSP47-loaded AA-T3A-C12 to silence HSP47, a therapeutic target of liver fibrosis, and compared it with the benchmark MC3 LNP. Notably, activated fibroblasts/HSCs were identified as the primary source of HSP47 in the fibrotic liver[8], which we also confirmed (Fig. S16). Immunofluorescence (IF) staining results showed that HSP47 expression in activated 3T3 fibroblasts was largely inhibited after AA-T3A-C12/siHSP47 LNP treatment (Figs. 4c and S17), and its effect was more potent than MC3/siHSP47 LNP. In contrast, AA-T3A-C12/siGFP LNP had no effect on the expression of HSP47. The superior silencing ability of AA-T3A-C12/siHSP47 LNP in activated 3T3 fibroblasts was further confirmed by Western blot analysis (Fig. 4d), as it mediated ~60% down-regulation of HSP47, which was much higher than the ~35% knockdown achieved by MC3/siHSP47 LNP. Afterward, their silencing activity was evaluated in activated primary HSCs, which were confirmed to express a much higher level of sigma receptors than hepatocytes (Fig. S18). The results showed that AA-T3A-C12/siHSP47 LNP achieved significantly higher knockdown efficiency compared to MC3/siHSP47 LNP (65% vs. 34%, Fig. 4e). Together, these results confirm that targeted delivery of siRNA into activated fibroblasts using AA-T3A-C12 LNP enables robust gene knockdown.

## Potent in vivo HSP47 silencing via AA-T3A-C12 LNP

After confirming potent in vitro HSP47 silencing, we then evaluated the in vivo performance and therapeutic potential of AA-T3A-C12/siHSP47 LNP. First, the biodistribution was investigated in fibrotic mice after tail vein injection of Cy5-siRNA-loaded LNPs. MC3 LNP was included as a positive control due to its well-known hepatic accumulation and transfection. As expected, MC3 LNP predominantly accumulated in the liver (Fig. 5a). AA-T3A-C12 LNP exhibited a similar organ distribution pattern compared to MC3 LNP with primary liver localization. Since there were no observable differences in liver accumulation between AA-T3A-C12 LNP and MC3 LNP ($p = 0.443$), they were compared for in vivo silencing activity and therapeutic efficacy. Non-targeted T3A-C12 LNP was not included in further studies due to its low siRNA EE and poor liver accumulation (Table S1 and Fig. S19). To further investigate the HSC-targeting ability of LNPs, livers were cryosectioned and α-smooth muscle actin (α-SMA, a marker of activated HSCs) was stained for immunofluorescence analysis. Confocal imaging results showed that more AA-T3A-C12 LNP co-localized with or were close to activated HSCs compared to MC3 LNP (Fig. S20), presumably due to the strong affinity between anisamide ligands and sigma receptors. Together, these results demonstrate the superiority of AA-T3A-C12 LNP to target the liver and activated HSCs.

Next, therapeutic studies were conducted in fibrotic mice that were treated with CCl₄ twice weekly for 4 weeks to induce liver fibrosis (Fig. 5b). Healthy mice were used as an untreated control (Group 1, G1). During the last two weeks of the experiment, fibrotic mice were treated twice weekly with PBS (G2), AA-T3A-C12/siGFP LNP (G3), AA-T3A-C12/siHSP47 LNP (G4) or MC3/siHSP47 LNP (G5). The siRNA dose was pre-determined to be 5 μg (0.2 mg/kg) for each injection, which was lower than the median effective dose (ED50 = 0.34 mg/kg) for AA-T3A-C12/siHSP47 LNP but still achieved moderate HSP47 knockdown (Fig. S21). Interestingly, although all CCl₄-treated mice suffered substantial weight loss compared to healthy mice at the beginning, fibrotic mice treated with AA-T3A-C12/siHSP47 LNP or MC3/siHSP47 LNP gradually gained body weight (Fig. 5c). At the end of the experiment, the body weight of AA-T3A-C12/siHSP47 LNP- or MC3/siHSP47 LNP-treated fibrotic mice was comparable to healthy mice (Fig. 5d). Additionally, AA-T3A-C12/siGFP LNP treatment did not cause additional weight loss compared to PBS treatment in fibrotic mice ($p = 0.907$). These results suggest that AA-T3A-C12 LNP is well-tolerated by mice, and that silencing of HSP47 (Fig. 5e and f) could potentially aid in overcoming liver fibrosis-mediated weight loss.

To observe the changes in HSP47 expression, liver samples from each group were sectioned for IF staining. Confocal imaging of liver

sections demonstrated that minimal HSP47 expression was observed in healthy mice, but CCl₄ treatment induced abundant expression of HSP47, especially around central veins (Fig. 5e). Encouragingly, AA-T3A-C12/siHSP47 LNP treatment significantly down-regulated HSP47 expression in fibrotic mice. In contrast, MC3/siHSP47 LNP treatment only moderately reduced HSP47 expression due to its predominant hepatocyte transfection[41]. Western blot analysis of HSP47 in liver lysates showed that minimal HSP47 was detected in healthy mice (Fig. 5f), but a significant increase in HSP47 was observed in fibrotic mice. However, HSP47 expression in fibrotic mice was largely suppressed by AA-T3A-C12/siHSP47 LNP treatment. Quantitative analysis of Western blot results indicated that AA-T3A-C12/siHSP47 LNP treatment led to a 65% knockdown of HSP47 compared to PBS treatment in fibrotic mice, which was >2-fold more potent than the 31% knockdown achieved by MC3/siHSP47 LNP. Together, these results demonstrate that potent HSP47 silencing can be achieved in fibrotic mice by treatment with AA-T3A-C12/siHSP47 LNP.

## Reduction of liver fibrosis after AA-T3A-C12 LNP-mediated HSP47 silencing

After confirming the superior knockdown of HSP47 by AA-T3A-C12/siHSP47 LNP, we then examined its anti-fibrotic efficacy. Livers harvested from different treatment groups were subjected to macroscopic and histopathological analysis. Livers from healthy mice had a regular and smooth surface, while livers from fibrotic mice had a rough surface due to CCl₄-induced liver damage (Fig. 6a). Encouragingly, treatment with AA-T3A-C12/siHSP47 LNP enhanced the recovery of the damaged liver, as it had a more smooth and normal appearance than other treatment groups. Hematoxylin and eosin (H&E) staining of healthy livers revealed normal histological architecture. CCl₄ treatment induced extensive neutrophil infiltration and irregular regenerating pseudolobules with dense fibrotic septa in the liver. Additionally, extensive contraction and death of hepatocytes were observed in CCl₄-treated mice. However, livers from AA-T3A-C12/siHSP47 LNP-treated fibrotic mice exhibited a more normal histological structure and had fewer fibrotic septa, as well as fewer apoptotic hepatocytes compared to other treatment groups. Liver sections were further stained with Picosirius red to evaluate collagen coverage and tissue fibrosis. Healthy livers revealed normal lobular architecture and minimal collagen fibers. As expected, CCl₄ treatment led to extensive collagen deposition and pseudolobular formation, but AA-T3A-C12/siHSP47 LNP treatment remarkably reduced collagen fibers and pseudolobules with minimal collagen staining around central veins. Morphometric quantification of Picosirius red stained areas further confirmed the significantly decreased collagen deposition in AA-T3A-C12/siHSP47 LNP-treated fibrotic mice, which was more significant than that of the MC3/siHSP47 LNP-treated group ($p < 0.001$, Fig. 6b). Together, these results suggest that targeted knockdown of HSP47 in activated HSCs enabled by AA-T3A-C12 LNP successfully reduce collagen deposition and alleviate liver fibrosis, with better anti-fibrotic efficacy than MC3 LNP.

## Safety evaluation

Finally, we conducted the safety evaluation of these treatment regimens. First, serum alanine aminotransferase (ALT), aspartate aminotransferase (AST), and total bilirubin (TBIL) were quantified to assess liver toxicity (Fig. 6c–e). ALT, AST, and TBIL markedly increased in fibrotic mice compared to healthy mice, as a result of CCl₄-induced liver injury. AA-T3A-C12/siGFP LNP treatment did not further increase ALT, AST, and TBIL levels, indicating that AA-T3A-C12 LNP was well-tolerated without exacerbating the liver injury. Moreover, AA-T3A-C12/siHSP47 LNP treatment slightly decreased ALT, AST, and TBIL levels in fibrotic mice, but these decreases were not significant. Since fibrotic mice were continuously insulted with CCl₄ (Fig. 5b), it is reasonable to expect that liver functions were not significantly improved despite the obvious attenuation of the fibrotic process after AA-T3A-C12/siHSP47

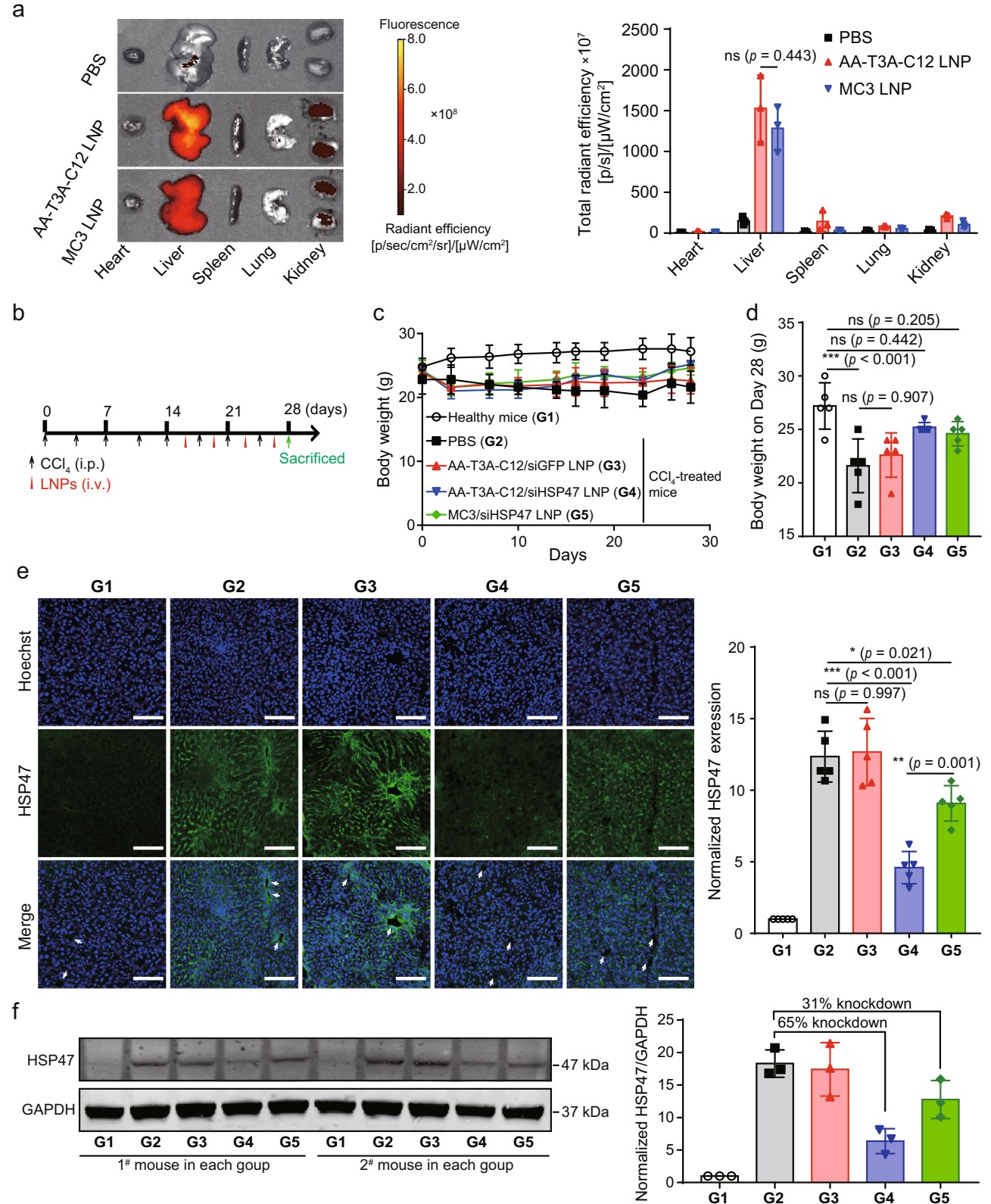

LNP treatment. Additionally, although CCl₄ treatment resulted in the elevation of proinflammatory cytokines including tumor necrosis factor (TNF)-α and interleukin (IL)−6 (Fig. S22), all LNP treatment regimens did not increase inflammation levels. Further, histopathological analysis of other organs including the heart, spleen, lung, and kidney was performed (Fig. S23). No histological differences were observed in these organs in all fibrotic mice groups compared to the healthy mice group, suggesting that neither CCl₄ nor LNPs induced noticeable damage to these organs. Together, these results demonstrate a good safety profile of AA-T3A-C12/siHSP47 LNP.

## Discussion

Liver fibrosis results from the excessive deposition of ECM, which is mainly produced by activated HSCs[5,6]. Previous studies have

**Fig. 5 | Biodistribution and HSP47 silencing activity of LNPs in fibrotic mice.**
**a** Ex vivo fluorescence imaging and signal quantification of major organs from PBS, AA-T3A-C12 LNP/Cy5-siRNA or MC3 LNP/Cy5-siRNA treated fibrotic mice (representative dataset from $n = 3$/group). **b** Scheme of $CCl_4$ and LNP treatment. Mice received intraperitoneal (i.p.) injections of 20% $CCl_4$ (0.7 μl/g) in corn oil twice a week for 4 weeks. LNPs were intravenously (i.v.) administered at a siRNA dose of 5 μg/mouse twice weekly for 2 weeks. **c** Body weight changes of mice over time during the experiment ($n = 5$/group). **d** Body weight at the end of the experiment ($n = 5$/group). **e** IF staining of HSP47 in liver sections (representative dataset from $n = 5$/group). Arrows indicate central veins. Quantitative analysis was performed

using ImageJ software ($n = 5$/group). Scale bar: 100 μm. **f** Western blot analysis of HSP47 expression in liver lysates (representative dataset from $n = 3$/group). GAPDH was used as an internal control. Representative images for two sets of mouse liver samples are shown. Quantitative analysis was performed using ImageJ software ($n = 3$/group). Data are presented as mean ± SD. G1, healthy mice; G2, PBS-treated fibrotic mice; G3, AA-T3A-C12/siGFP LNP-treated fibrotic mice; G4, AA-T3A-C12/siHSP47 LNP-treated fibrotic mice; G5, MC3/siHSP47 LNP-treated fibrotic mice. ns, not significant; *$p < 0.05$; **$p < 0.01$; ***$p < 0.001$. **a, d, e** one-way ANOVA with Tukey's correction. Source data are provided as a Source Data file.

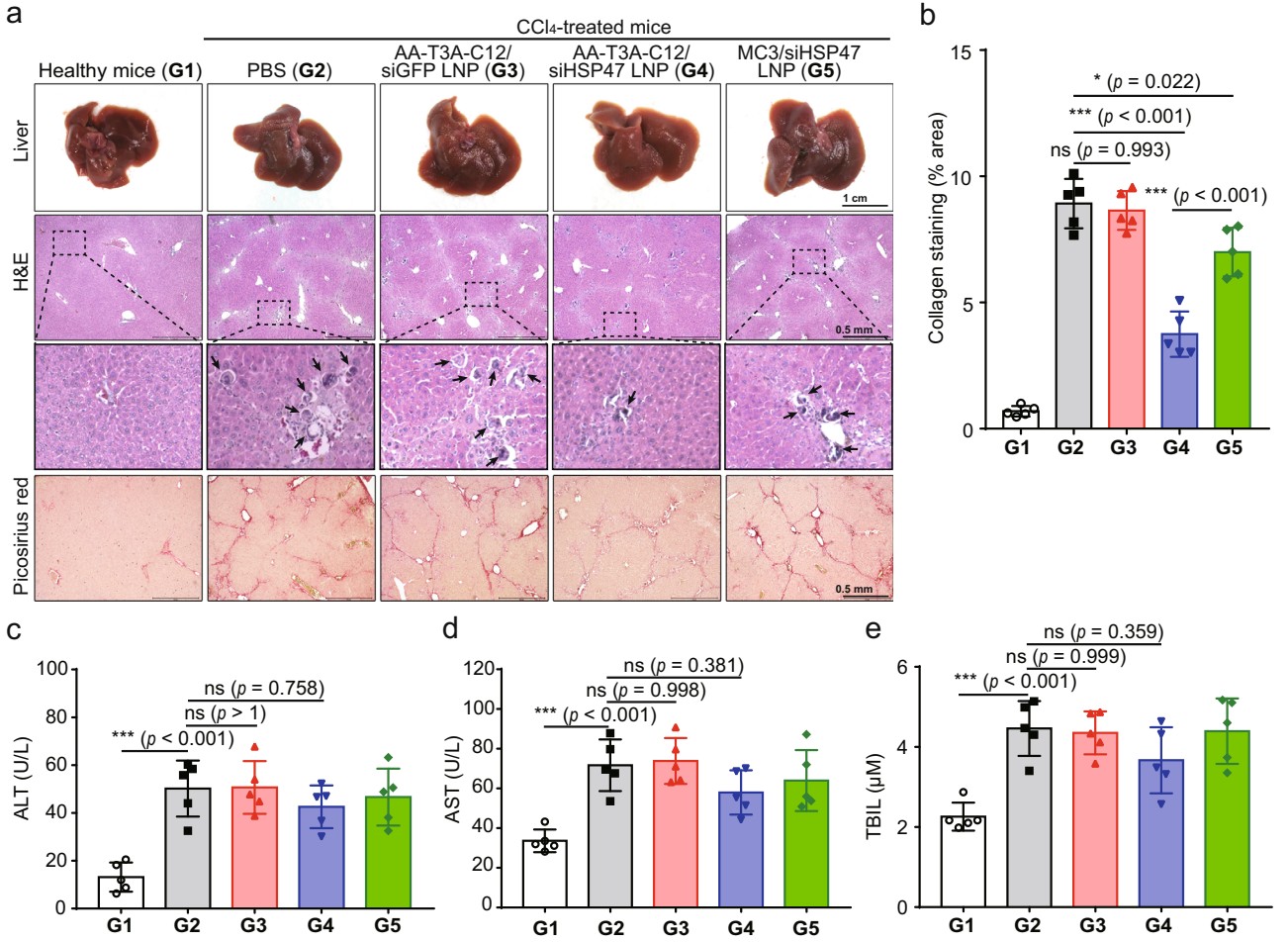

**Fig. 6 | Macroscopic, histopathological, and biochemical analysis of liver fibrosis. a** Representative images of livers and liver sections stained with hematoxylin and eosin (H&E) or Picrosirius red (representative dataset from $n = 5$/group). Black arrows indicate apoptotic hepatocytes. **b** Morphometric quantification of Picrosirius red stained areas by ImageJ software ($n = 5$/group).

**c** Quantification of serum alanine aminotransferase (ALT, $n = 5$/group).
**d** Quantification of serum aspartate aminotransferase (AST, $n = 5$/group).
**e** Quantification of serum total bilirubin (TBIL, $n = 5$/group). Data are presented as mean ± SD. ns not significant; *$p < 0.05$; ***$p < 0.001$. **b–e** one-way ANOVA with Tukey's correction. Source data are provided as a Source Data file.

demonstrated that targeted knockdown of HSP47 in activated HSCs by siHSP47-loaded, vitamin A-coupled cationic liposomes (namely ND-L02-s0201) can reduce collagen production and resolve liver fibrosis[9], which resulted in a clinical trial (NCT02227459). However, cationic liposomes are generally associated with significant toxicity and immunogenicity[13]. In contrast, ionizable LNPs are neutral at physiological pH and only become positively charged at acidic pH, which largely reduces toxicity and non-specific interactions with blood components[13,15,42,43]. Therefore, they represent the most clinically advanced non-viral vector, especially for hepatic gene delivery, with one FDA-approved siRNA-MC3 LNP drug, patisiran. Interestingly, the lipidoid greatly affects LNP tropism, and LNPs comprised of different

lipidoids have been reported to potentially transfect hepatocytes, LSECs, and/or Kupffer cells[22–26].

Since no lipidoid has been reported to actively target activated HSCs, we hypothesized that lipidoids with an HSC-targeting motif could intrinsically mediate LNP delivery to these cells (Fig. 1b). Therefore, we selected a small molecule targeting ligand anisamide as the building block and developed a one-pot, two-step modular synthetic method to enable fast and parallel synthesis of a combinatorial library of AA-lipidoids (Fig. 2a). It is worth noting that the amine−succinimide coupling reaction and ring-opening reaction can be combined due to their simplicity, compatibility and high efficiency, which greatly simplifies synthesis and reduces workload. Afterward,

we used a two-round screening strategy with a first-round screening of silencing efficiency and a second-round screening of selectivity in activated fibroblasts, from which a potent and targeted AA-lipidoid (AA-T3A-C12) was identified (Fig. 2).

Systematic in vitro cellular uptake (Figs. 3 and S12, 13) and gene knockdown (Fig. 4) experiments confirmed that AA-T3A-C12 LNP mediated greater RNA delivery and transfection of activated fibroblasts than non-targeted T3A-C12 LNP and MC3 LNP. Moreover, AA-T3A-C12 LNP was predominantly sequestered and taken up by perivascular HSCs after systemic delivery (Fig. S20). Interestingly, in a mouse model of CCl₄-induced liver fibrosis, siHSP47-loaded AA-T3A-C12 LNP also remarkably outperformed MC3 LNP in silencing HSP47 (65% vs. 31%), reducing collagen deposition and alleviating liver fibrosis despite their comparable liver accumulation (Figs. 5 and 6a, b). Finally, we showed that the AA-T3A-C12 LNP itself was well-tolerated by mice and did not exacerbate CCl₄-induced liver damage (Fig. 6c–e).

There are also some limitations of this study. First, to enable high-throughput screening of LNPs, 3T3 fibroblasts with the GFP reporter genes were chosen instead of liver-resident HSCs. Nevertheless, 3T3 cells have been widely used as a model fibroblast cell line to study various fibrotic diseases[33,44]. Moreover, stimulation of 3T3 cells with TGF-β can induce the overexpression of sigma receptors[33], which mimics activated HSCs with overexpressed sigma receptors in the fibrotic liver[32,34]. Second, since the conjugation of anisamide changes the structure and hydrophobicity of the lipidoid, some physical properties of LNPs could be changed as well, making strict comparisons to their lipidoid analogs without anisamide difficult to conduct. For example, unlike AA-T3A-C12 LNP, T3A-C12 LNP showed low siRNA EE and poor liver accumulation, which makes it unsuitable for in vivo studies. The lower solubility of T3A-C12, which has six epoxide tails compared to AA-T3A-C12 which has four, could be a potential cause of its poor performance. Nevertheless, we have confirmed that the potency of AA-lipidoids was highly dependent on sigma receptors based on the analysis of the structure–activity relationships, which was not observed for lipidoids without anisamide. It is worth mentioning that cell-specific targeting is also dependent on passive targeting[45]. Only when the liver is reached can LNPs bind to HSCs and be internalized. Therefore, both passive targeting and active targeting are important criteria for LNP-mediated RNA delivery to activated HSCs, which was only achieved by AA-T3A-C12 LNP.

In summary, a combinatorial library of anisamide-tethered lipidoids was prepared using a one-pot, two-step modular synthetic method. By adopting a two-step screening strategy, the AA-T3A-C12 lipidoid with both high potency and selectivity was successfully discovered, which mediated robust RNA delivery and gene silencing in activated fibroblasts. In a preclinical model of liver fibrosis, targeted delivery of HSP47 siRNA by AA-T3A-C12 LNP led to ~65% silencing of HSP47, which outperformed the benchmark MC3 LNP by two-fold. The significant reduction of collagen deposition and alleviation of liver fibrosis without noticeable toxicity achieved by AA-T3A-C12/siHSP47 LNP suggests the potential of this targeted LNP platform for anti-fibrotic therapy. Furthermore, the synthetic and screening methodologies developed here can potentially be utilized to synthesize and identify new ligand-tethered lipidoids that can achieve potent and targeted gene delivery to cell and tissue types that are difficult to access using traditional LNP technologies.

## Methods

### Materials
Core 200 was customized from Enamine (Monmouth Junction, NJ) and other polyamine cores were purchased from Sigma Aldrich, Tokyo Chemical Industry (TCI), and Alfa Aesar. Epoxydodecane (C12), epoxytetradecane (C14), epoxyhexadecane (C16), 4-methoxybenzoic acid, N,N′-Dicyclohexylcarbodiimide (DCC), N-Hydroxysuccinimide (NHS), HSP47 siRNA pool (NM_001111043, NM_001111044, and NM_009825)

and Cy5-siRNA were purchased from Sigma Aldrich. GFP siRNA (cat. P-002048-01-50) and DharmaFECT transfection reagents were purchased from Horizon Discovery Ltd. 1% agarose gels with SYBR™ Safe (#A42100) were purchased from ThermoFisher. Recombinant mouse TGF-β1 (cat. 7666-MB) was obtained from R&D. DSPC (#850365), cholesterol (#700100), and C14-PEG2000 (#880150) were bought from Avanti Polar Lipids. DLin-MC3-DMA was purchased from Med-Chem Express (Monmouth Junction, NJ). Luciferase mRNA was produced through an in vitro transcription (IVT) method[46].

### Synthesis of Anisoyl-NHS ester
4-methoxybenzoic acid (960 mg, 6.4 mmol) and NHS (800 mg, 7 mmol) were dissolved in 36 mL DCM and stirred at 0 °C. DCC (1440 mg, 7 mmol) in 40 mL DCM was added drop-wise for 30 min. The resulting mixture was further stirred at 0 °C for 1 h and then placed in a refrigerator at 0 °C overnight. The precipitated solid was removed by filtration, and the filtrate was dried to give a crude product. Recrystallization of the crude solid from 2-propanol afforded 1.2 g of Anisoyl-NHS ester (yield 75%). Anisoyl-NHS ester was characterized by mass spectrometry (Fig. S1) and nuclear magnetic resonance spectroscopy (Fig. S2).

### General method for the synthesis of anisamide-tethered lipidoids
AA-lipidoids were synthesized using a one-pot, two-step method. First, polyamine (1 equiv.) and anisoyl-NHS ester (1 equiv.) were combined in EtOH, and TEA (1.2 equiv.) was then added. The resulting mixture was mildly heated (30–50 °C) for 2 h, in which, anisoyl-NHS ester was slowly dissolved and reacted with a polyamine. Afterward, excessive alkyl epoxide (e.g., 4.8 equiv. for T3A core) was added and the mixture was heated at 80 °C for 2 days[35,47]. Crude products were used for initial in vitro screening. The top-performing AA-lipidoid, AA-T3A-C12 was purified by a CombiFlash Nextgen 300+ chromatography system (Teledyne ISCO) with gradient elution from CH₂Cl₂ to 75:22:3 CH₂Cl₂/MeOH/NH₄OH (aq). The target fraction was identified by mass spectrometry (Fig. S6) and nuclear magnetic resonance spectroscopy (Fig. S7).

### General method for the synthesis of lipidoids without anisamide
Lipidoids without anisamide were synthesized by reacting excessive alkyl epoxides (e.g., 7.2 equiv. for T3A core) with polyamines at 80 °C for 2 days[35,47].

### LNP formulation
An organic phase was prepared in ethanol by solubilizing ionizable lipid (lipidoid, AA-lipidoid, or MC3), DSPC, cholesterol, and C14-PEG2000 at a molar ratio of 50:10:38.5:1.5. The aqueous phase was prepared in 10 mM citrate buffer (pH 3) with siRNA or luciferase mRNA. The aqueous phase and organic phase were mixed at an ionizable lipid:RNA weight ratio of 10:1 and at a flow rate of 1.8 mL/min and 0.6 mL/min (3:1) using Pump 33 DDS syringe pumps (Harvard Apparatus, MA) in a microfluidic device with a staggered herringbone micromixer design. The microfluidic devices were fabricated in polydimethylsiloxane according to standard soft lithographic procedures[37]. A two-step exposure process was used to create the SU-8 master with positive channel features on a silicon wafer. Each mixing channel is 4 cm in length. LNPs were dialyzed against 1 × PBS in a 20 kDa MWCO cassette for 2 h, filtered through a 0.22 μM filter, and stored at 4 °C. For in vitro screening, LNPs were prepared by pipette mixing and directly used to treat cells without further dialysis.

### LNP characterization
The hydrodynamic diameter, polydispersity index (PDI), and zeta potential of LNPs were measured using a Zetasizer Nano ZS90 (Malvern Instruments, Malvern, UK). The morphology of LNPs were

characterized by transmission electron microscopy (JEOL 1010, Tokyo, Japan). The siRNA–LNP complex was analyzed by agarose gel electrophoresis[48]. siRNA encapsulation efficiency was determined using a modified Quant-iT RiboGreen RNA assay (Invitrogen)[47]. The p$K_a$ of LNP was determined using a 6-(p-toluidinyl)naphthalene-2-sulfonic acid (TNS) assay[49].

## Cell culture and animal studies

The murine NIH 3T3 fibroblast cell line was kindly provided by Prof. Andrew Tsourkas (University of Pennsylvania), who obtained it from the American Type Culture Collection ATCC (cat. CRL-1658). The NIH 3T3-GFP cell line was purchased from Cell Biolabs (#AKR-214). The primary murine HSCs were obtained from Sciencell (cat. M5300-57). The immortalized primary murine H2.35 hepatocyte cell line was kindly provided by Prof. James M. Wilson (University of Pennsylvania), who obtained it from ATCC (cat. CRL-1995). The murine bEnd.3 endothelial cell line (cat. CRL-2299) and the murine RAW264.7 macrophage cell line (cat. TIB-71) were obtained from the ATCC. Primary HSCs were maintained in Stellate Cell Medium (#5301, Sciencell) and used between passages 3 and 8. All other cells were maintained in Dulbecco's modified Eagle medium (DMEM) supplemented with 10% fetal bovine serum (FBS), 100 U/mL penicillin, and 100 μg/mL streptomycin at 37 °C in a humidified incubator of 5% $CO_2$, and routinely tested for mycoplasma contamination. Activated 3T3 fibroblasts and HSCs were obtained by stimulation with 10 ng/ml of TGF-β for 24 h.

All animal protocols were approved by the Institutional Animal Care and Use Committee (IACUC) of the University of Pennsylvania (No. 806540), and animal procedures were performed in accordance with the Guidelines for Care and Use of Laboratory Animals at the University of Pennsylvania. Balb/c male mice (6–8 weeks age, 22–25 g body weight) were purchased from Jackson Laboratory and housed in a specific-pathogen-free animal facility at ambient temperature (22 ± 2 °C), air humidity 40–70% and 12-h dark/12-h light cycle. Liver fibrosis was induced by intraperitoneal (i.p.) injection of 20% $CCl_4$ (0.7 μl/g) in corn oil twice a week for 4 weeks.

## High-throughput screening of LNPs in vitro

High-throughput screening of LNPs was performed by evaluating GFP knockdown efficiency in 96-well plates using a plate reader assay. 3T3-GFP cells were re-suspended in DMEM containing 10 ng/ml of TGF-β and seeded into a 96-well plate at a density of 5,000 cells per well. After 24 h, activated cells were treated with various LNPs loaded with siGFP (50 nM). At 48 h post-treatment, the GFP signal in each well was obtained using an Infinite M Plex plate reader (Tecan, Morrisville, NC, USA) with an excitation of 488 nm and an emission of 520 nm. Cell viability was measured using a CellTiter-Glo Luminescent Cell Viability Assay according to the manufacturer's instruction (G7572, Promega).

The top-performing LNPs were subjected to the second round of screening by evaluating their dependency on sigma receptor-mediated transfection. Activated 3T3-GFP cells were treated with or without haloperidol (30 μM) for 2 h before various LNPs loaded with siGFP (50 nM) were used to treat these cells for another 48 h. The GFP signal in each well was obtained as described above.

## Cellular uptake and mRNA transfection

Activated 3T3-GFP cells in a six-well plate (5 × 10⁵ per well) were pre-treated with or without haloperidol (30 μM) for 2 h before treatment with AA-T3A-C12 LNP loaded with Cy5-siRNA (50 nM). At 6 h post-treatment, cells were collected for analysis using flow cytometry (BD, LSR II).

3T3-GFP cells (2.5 × 10⁵ per well) and H2.35 cells (2.5 × 10⁵ per well) were co-cultured in a 6-well plate and stimulated with TGF-β for 24 h. Cells were treated with AA-T3A-C12 LNP, T3A-C12 LNP or MC3 LNP loaded with Cy5-siRNA (50 nM) for 6 h before collection for flow cytometry analysis.

Activated 3T3-GFP cells or H2.35 cells in a 96-well plate (5000 per well) were treated with AA-T3A-C12 LNP or MC3 LNP loaded with luciferase mRNA (15 ng/well) for 24 h. Luciferase expression was evaluated by Luciferase Reporter 1000 Assay System according to the manufacturer's protocol (E4550, Promega), and cell viability was measured using a CellTiter-Glo Luminescent Cell Viability Assay.

## Dose- and time-independent GFP knockdown in vitro

Activated 3T3-GFP cells in a 96-well plate (5000 per well) were treated with AA-T3A-C12/siGFP LNP at siRNA concentrations ranging from 25 to 100 nM. GFP expression was evaluated by a plate reader at 24 and 48 h post-transfection. Cell viability was measured as described above.

To evaluate GFP knockdown efficiency using flow cytometry, activated 3T3-GFP cells in a six-well plate (5 × 10⁵ per well) were treated with AA-T3A-C12/siGFP LNP at siRNA concentrations ranging from 25 to 100 nM for 48 h. Cells were collected for analysis by flow cytometry.

## Western blot analysis

20 μg of cell protein was loaded for electrophoresis. Blots were incubated with HSP47 antibody (1:1000, #NBP1-97491, Novus Biologicals) or Sigma Receptor antibody (1:200, #sc-137075, Santa Cruz) overnight at 4 °C. After being washed three times, blots were incubated with IRDye 800CW donkey anti-mouse IgG secondary antibody (#925-32212, LiCor) at a 1:10,000 dilution for 1 h at RT. Blots were imaged using an Odyssey IR Imaging System. GAPDH was used an internal control.

## HSP47 knockdown in vitro

Activated 3T3 cells in 35-mm glass-bottom dishes were treated with AA-T3A-C12/siGFP LNP, AA-T3A-C12/siHSP47 LNP or MC3/siHSP47 LNP (50 nM). 48 h later, cells were subjected to immunofluorescence staining of HSP47 using an Immunofluorescence Application Solutions Kit (#12727, CST). Samples were incubated with HSP47 antibody at a 1:200 dilution overnight. After being washed three times, samples were incubated with Alexa Fluor® 488 conjugated goat anti-mouse IgG (H + L), F(ab')₂ fragment antibody (#4408, CST) at a 1:1000 dilution for 1 h. Nuclei were stained with Hoechst 33342 (10 μg/mL) before images were taken using a confocal laser scanning microscope (LSM 710, Zeiss).

To evaluate HSP47 knockdown by Western blot, activated 3T3 cells or HSCs in a six-well plate (5 × 10⁵ per well) were treated with AA-T3A-C12/siGFP LNP, AA-T3A-C12/siHSP47 LNP, or MC3/siHSP47 LNP (50 nM) for 48 h. Cells were harvested for Western blot analysis.

## LNP biodistribution

AA-T3A-C12 LNP or MC3 LNP loaded with Cy5-siRNA (5 μg/mouse) was i.v. injected into each fibrotic mouse (n = 3). 1 h post-injection, mice were euthanized and major organs were harvested for ex vivo imaging using an in vivo imaging system (IVIS Lumina 3, PerkinElmer). Livers were collected to prepare 10 μm cryosections. Samples were stained with FITC-conjugated α-smooth muscle actin antibody (#F3777, Sigma) at a 1:500 dilution overnight at 4 °C. Nuclei were stained with Hoechst 33342 (10 μg/mL) before images were taken using a confocal laser scanning microscope.

## Therapeutic studies

Mice were treated with $CCl_4$ twice weekly for 4 weeks. On week 3, mice (n = 5) were treated with PBS, AA-T3A-C12/siGFP LNP, AA-T3A-C12/siHSP47 LNP or MC3/siHSP47 LNP (5 μg siRNA/mouse) twice weekly for 2 weeks. Untreated (healthy) mice (n = 5) were used as a control group. Body weight was recorded twice a week during the experiment. Two days after the last treatment, mice were anesthetized and blood was collected through the retro-orbital route. Serum samples were prepared and stored at −80 °C

until use. Major organs (heart, liver, spleen, lung, and kidney) were collected. A part of the liver was collected for cryosectioning and subjected to immunofluorescence staining of HSP47 as described above. A part of the liver was homogenized in RIPA buffer (Thermo Scientific) and proteins were extracted for Western blot analysis of HSP47 expression as described above.

### Histological and blood biochemical analysis

Major organs were fixed in 4% paraformaldehyde, embedded in paraffin, cut into 5 μm sections, and stained with hematoxylin and eosin (H&E) for pathological analysis. Liver sections were also stained with Picrosirius Red (ab150681, Abcam). Images were taken using a microscope (FL Auto 2 Imaging System, EVOS).

Liver toxicities were evaluated by measuring aspartate aminotransferase (#701640, Cayman), alanine aminotransferase (#700260, Cayman), and total bilirubin (#701720, Cayman) in the serum. The immunotoxicity was evaluated by measuring serum IL-6 (#88-7064, Invitrogen) and TNF-α (#88-7324, Invitrogen) using enzyme-linked immunosorbent assay.

### Statistics and reproducibility

Data are presented as mean ± SD. Student's $t$-test or one-way analysis of variance (ANOVA) followed by the Tukey test was applied for comparison between two groups or among multiple groups using Graphpad Prism 7.0, respectively. $p < 0.05$ was considered to be statistically significant. Each experiment is repeated at least three times independently with similar results and the representative dataset is presented.

### Reporting summary

Further information on research design is available in the Nature Portfolio Reporting Summary linked to this article.

## Data availability

All relevant data supporting the key findings of this study are available within the article and its Supplementary Information files or from the corresponding author upon reasonable request. Source data are provided with this paper.

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

## Acknowledgements

M.J.M. acknowledges support from a US National Institutes of Health (NIH) Director's New Innovator Award (DP2 TR002776), a Burroughs Wellcome Fund Career Award at the Scientific Interface (CASI), a US National Science Foundation CAREER award (CBET-2145491), and additional funding from the National Institutes of Health (NCI R01 CA241661, NCI R37 CA244911, and NIDDK R01 DK123049). S.J.S. is supported by an NSF Graduate Research Fellowship (Award 1845298). The authors acknowledge Christian Figueroa-Espada from the University of Pennsylvania for his help in figure creation using BioRender.com.

## Author contributions

Conceptualization, X.H. and M.J.M.; Methodology, X.H., N.G., L.X., R.E., and M.M.B., Investigation, X.H., and N.G.; Visualization, X.H. and S.J.S.; Supervision, M.J.M.; Writing—original draft, X.H. and N.G.; Writing—review & editing, R.E., M.-G.A., D.W., and M.J.M.; Funding acquisition, M.J.M.

## Competing interests

X.H. and M.J.M. are inventors on a patent filed by the Trustees of the University of Pennsylvania (U.S. Provisional Patent Application No. 63/286,760, filed December 7, 2021) describing the ligand-tethered lipid nanoparticle technology in this manuscript. All other authors declare they have no competing interests.
