## [Peer Review File · Nature Communications]

Ligand-Tethered Lipid Nanoparticles for Targeted RNA Delivery to Treat Liver FibrosisREVIEWER COMMENTS

Reviewer #1 (Remarks to the Author):

This is an interesting study on targeting the cell type that drives liver fibrosis, the hepatic stellate cell. The authors generated lipid nanoparticles in which the state of the art ionizable lipid (MC3) is replaced by a novel lipidoid.

1) I would not call the particles Ligand-Tethered Lipidoids, but rather lipid nanoparticles.

2) The authors achieve enhanced targeting to the HSC. However, the purest and clearest evidence that their novel siRNA vector works would be if they isolate HSC from the livers of healthy or ideally fibrotic mice and then demonstrate a knockdown of HSP47 in HSC, but not other liver cell types (Kupffer cells, endothelial cells, hepatocytes).

3) Figure 5A: I wonder if the signals that are shown are really signals from the LNP. An untreated control is missing. I know from my own experiences that using the IVIS imaging device, also untreated livers appear red - due to the autofluorescence of the liver. Thus, an untreated group of mice must be added here. Assure that you use the same IVIS settings again.

4) A comparison of their formulation to the one from Sato et al 2008, Nature Biotechnology, in which they used Vitamin A-loaded nanoparticles would, to my mind, be a better comparison than towards MC3-LNP. MC3 is optimized for a release in hepatocytes and not for stellate cells. Therefore, it can not be a general benchmark for knockdown, only for a knockdown in hepatocytes.

Reviewer #2 (Remarks to the Author):

This is a well written manuscript. The authors have generated a novel small molecule tethered LNP that delivers really potent siRNA based system as compared to the current gold standard. The authors elegantly show silencing and functional knock down in liver hepatic stellate cells. Targeting HSCs has remained a challenge and this work shows it is possible to target this cell population. I recommend publication with minor revisions.

1. Is there an effect of anisamide as a small molecule in fibrosis. Will treatment in even cell types show that anisamide alone does not affect the fibrotic events. One experiment that uses anisamide in MC3 formulations or separately to show that effects are due to silencing and not an effect of anisamide in the formulation might improve the impact.

2. Is there a dose dependence in silencing. IC50 for silencing might be helpful or is the max silencing that can be achieved is limited to 65%.

3. Fig 5 legends is unclear with the numerical numbers.

4. Overall, this is a well written work.

Reviewer #3 (Remarks to the Author):

The manuscript entitled Ligand-Tethered Lipidoids for Targeted RNA Delivery to Treat Liver Fibrosis by Han et. al developed a combinatorial library of anisamide ligand-tethered lipidoids (AA-lipidoids) to identify AA-lipidoids with both high potency and selectivity to deliver RNA payloads to activated fibroblasts. In a preclinical model of liver fibrosis, AA-lipidoids reduced collagen deposition and liver fibrosis. Most experimental evidence is supported with proper experimental data. But still, the manuscript lacks in few portions which the authors need to address before its acceptance for publication in a high-quality journal like Nature Communications. Below are the Major comments that the authors have to address to improve the overall quality of the manuscript.

1. Detailed information on the microfluidic chip was not mentioned in this manuscript. To improve the transparency of the work, the design of microfluidic chips should be provided. Main experimental parameters, such as the flow rate, are needed to be described.

2. How is the optimal ratio of nucleic acid to LNP determined? This important parameter must be identified by agarose gel electrophoresis (Nano Letters, 2018, 18(6): 3328-3334.).

3. The characterization of the nanoparticles is insufficient. For example, TEM images of LNP vector without DNA encapsulation should be added. Stability tests of LNPs in protein, as well as in serum, were completely missing.

4. The difficulty of gene transfection in primary cells (e.g. 3T3 cells and H2.35 cells) is significantly higher than that in cell lines. The efficacy of LNP should be tested in primary hepatocytes.

5. The efficacy of LNP was not comprehensive as evidenced by body weight data and immunofluorescence staining alone. More intuitive and convincing experiments such as ICG clearance tests, blood AST, blood ALT, and blood TBIL should be provided.
6. The contraction and death of hepatocytes are important indicators of liver fibrosis. Images of H&E sections in Fig. 6 should be enlarged to observe changes in cell morphology.
7. The use of symbols in manuscripts is very casual. For example, throughout the manuscript, the multiplication sign is written as the letter x. The authors should carefully check the paper.
8. Authors claimed that LNP was subsequently uptake by perivascular HSCs (Page 10, line 13). However, immunofluorescence did not differentiate between LNP phagocytosis by HSCs and hepatic macrophages. More evidence is needed to prove this claim.

More evidence is needed to prove the claim.

Reviewer #4 (Remarks to the Author):

Remarks to the authors:

Han and his team have done a good job in developing a ligand-targeted LNP-siRNA delivery system for a potential liver fibrosis gene therapy. In their research, an efficient 2-step in vitro LNP screening method was also established to contribute to the potential fast-track development of clinical and pharmaceutical applications. The highlight of their research is to report the noteworthy efficacy of the candidate AA-T3A-C12 LNP in the knockdown of the disease-related overexpressing HSP47 protein in liver fibroblasts. The manuscript was well-written with a thorough literature review citing the most important recent papers in LNP and LNP-targeting delivery. A significant amount of work was conducted, including synthesis, in vitro screening and in vivo biodistribution, efficacy and safety studies. Overall, the research is thoughtful and insightful, although some experiments need to be adjusted and added.

Major comments:

For the in vitro screening part, it was mentioned in the discussion that the 3T3 cell line was chosen because it expresses the sigma receptors. However, no evidence showed that the sigma reporters are also overexpressed in primary HSCs. Therefore, it might not show the relevance of settling the dependency on sigma receptor-mediated transfection efficiency as the round 2 selection, unless it was meant to be the indicator for 3T3/hepatic cells selectivity of the LNP expression only. Although the 3T3/hepatic cells co-culture results were impressive, it still could not support the conclusion that the AA-T3A-C12 LNP would be the best formulation in vivo based on the in vitro studies if it was settled like

this. The statement of “both passive and active targeting are important criteria” for LNP delivery could not be made if this has not been confirmed in primary liver fibroblasts. Addition of the primary cell line data of liver fibroblasts, or replacing the 3T3 cell line with primary liver fibroblasts showing the expression of sigma receptors (e.g. western blot), would be a more suitable model for this experiment.

Similarly, it appears that the feature of high dependency on sigma receptor-mediated transfection efficiency was chosen as the key and the only indication for fibroblast/hepatocyte selectivity in the comparison of AA-T3A-C12 and MC3 LNPs. However, no data was shown as to whether this feature may also impact the cellular uptake of MC3 LNP. This might not be a fair comparison to compare the ligand-targeted nanoparticle (AA-T3A-C12 LNP) with the particles following passive diffusion (MC3 LNP). More investigation on AA ligand-targeted MC3 LNP would be required if it is meant to prove that the reported lipid is more efficient than MC3. Also, it would make more sense if the MC3 LNP was also shown in the 2-step screening process as a control. It could be much easier to see the differences between AA-T3A-C12 and MC3 LNPs if the left Y-axis (MFI of siRNA) can be modified (e.g. % or 0 to 1 measurement/control) in some figures.

Minor comments:

In the Introduction, it was said that “However, to the best of our knowledge, lipidoids that intrinsically mediate efficient RNA delivery into activated HSCs have not been reported”. Actually, there are few papers that have been published on this topic, such as Specific hepatic delivery of procollagen α 1(I) siRNA in lipid-like nanoparticles resolves liver fibrosis (doi: 10.1002/hep.27936) and Comparison of DLin-MC3-DMA and ALC-0315 for siRNA Delivery to Hepatocytes and Hepatic Stellate Cells (doi: <https://doi.org/10.1021/acs.molpharmaceut.2c00033>). It would be beneficial to add these papers to the references, especially the second one to support the comparison with MC3 LNPs reported in the manuscript.

The authors should clarify if the formulation of the MC3 LNP that has been used for comparison is the standard formulation (MC3/DSPC/cholesterol/DMG-PEG= 50/10/38.5/1.5, N/P 6). It is noticeable that the reported lipid has a giant headgroup, which might lead to some issues of low efficiency of siRNA loading. This could be the reason for the slightly adjusted LNP formulation and the N/P ratio 10 of the AA-T3A-C12 LNP. It may lead to unfair comparison without notification of the formulation MC3 LNP, as the formulation can make considerable differences in transfection efficiency, gene expression levels and toxicity.

The LNP manufacturing method may raise concerns about the pH values of the LNP samples after 2 hours of dialysis. In previous research, it mentioned that 4- 6 hours of dialysis was essential to be performed to alter the pH of the samples from 4 to 7.4 instead of 2 hours from pH 3 to 7.4 in this

research. The pH of LNP samples may also impact the in vitro and in vivo results. Did the authors measure the pH of the LNP samples after dialysis?

REVIEWERS' COMMENTS AND AUTHORS' ANSWERS

Note: Our responses (standard typeface) to reviewers' comments (bold); the yellow highlighted words and sentences have been added to the main text.

Thank you very much for the opportunity to revise our manuscript. We very much appreciate your review of our manuscript. We thank all the reviewers for their insightful comments. We have revised the paper in accordance with the reviewers' suggestions and the editor's instructions, which have significantly improved our manuscript.

Specifically, we gathered extensive experimental evidence to support our claims. First, we formulated and tested a previously reported vitamin A-loaded liposome formulation for comparison as suggested by Reviewer #1. Second, more characterization of LNPs, including empty LNPs and the stability of LNPs in protein- or serum-supplemented solution, was conducted as suggested by Reviewer #3. Third, the HSP47 knockdown was further verified in primary HSCs as suggested by Reviewers #1 and #3. Fourth, additional methodological details were added.

All the consequential changes are highlighted in yellow in our resubmission. Specific discussion of the reviewers' concerns is present in our point-by-point response below.

Reviewer #1 (Remarks to the Author):

This is an interesting study on targeting the cell type that drives liver fibrosis, the hepatic stellate cell. The authors generated lipid nanoparticles in which the state of the art ionizable lipid (MC3) is replaced by a novel lipidoid.

Response: We thank the reviewer for their positive feedback of our manuscript, as well as for their time and effort in helping us improve the manuscript.

1) I would not call the particles Ligand-Tethered Lipidoids, but rather lipid nanoparticles.

Response 1: We thank the reviewer for this suggestion. We have now revised the manuscript title to "Ligand-Tethered **Lipid Nanoparticles** for Targeted RNA Delivery to Treat Liver Fibrosis".

2) The authors achieve enhanced targeting to the HSC. However, the purest and clearest evidence that their novel siRNA vector works would be if they isolate HSC from the livers of healthy or ideally fibrotic mice and then demonstrate a knockdown of HSP47 in HSC, but not other liver cell types (Kupffer cells, endothelial cells, hepatocytes).

Response 2: We thank the reviewer for this important suggestion. In this study, we developed a combinatorial library of AA-lipidoids and identified AA-T3A-C12 as a lipidoid that is highly efficient in delivering RNA payloads to activated fibroblasts by means of the strong affinity between anisamide and sigma receptors. We did not claim that AA-T3A-C12 LNP could achieve exclusive siRNA delivery to activated fibroblasts/HSCs and no other cell types. In fact, more LNPs were taken up by hepatocytes compared to fibroblasts in our fibroblast/hepatocyte co-culture experiment (**Fig. 3C**). However, we demonstrated that AA-T3A-C12 LNP greatly enhanced siRNA delivery to activated fibroblasts compared to T3A-C12 LNP or MC3 LNP despite their comparable delivery to hepatocytes (**Fig. 3C** and **S13**). Moreover, since activated HSCs are identified as the primary source of HSP47 in the fibrotic liver (Lab. Invest. 2005, 85, 789-797), the significant downregulation of HSP47 in the liver by our AA-T3A-C12/siHSP47 LNP is primarily attributed to the strong HSP47 knockdown in activated HSCs, despite that HSP47 knockdown could possibly be achieved in other cell types.

We have now acquired primary HSCs and compared the HSP47 expression in primary HSCs, 3T3 fibroblasts, immortalized primary hepatocytes H2.35 and other cell types (**Fig. S16**). HSP47 expression was dramatically higher in fibroblasts (primary HSCs and 3T3 cells) than other cell types. Specifically, hepatocytes and macrophages displayed lower expression of HSP47 compared to fibroblasts. Moreover, activated primary HSCs and 3T3 cells further up-regulate HSP47 expression. These results provide a rationale to target this effector cell population instead of other cells in the liver. Therefore, even if HSP47 knockdown is achieved in hepatocytes, macrophages, and endothelial cells but not in HSCs, the overall HSP47 expression in the liver cannot be greatly reduced and thus liver fibrosis cannot be reduced. The corresponding discussion has now been added to the manuscript (Page 9, line 255) as follows: “Notably, activated fibroblasts/HSCs are identified as the primary source of HSP47 in the fibrotic liver⁸, which was also confirmed by us (**Fig. S16**).”

Fig S16. Western blot analysis of HSP47 expression in different cell types. Activated 3T3 cells and HSCs were obtained by stimulation with 10 ng/ml of TGF- β for 24 h. GAPDH was used as an internal control. HSP47 expression was normalized to endothelial cells. HSP47 expression was much higher in fibroblasts (primary HSCs and 3T3 cells) than other cell types. Moreover, activated fibroblasts further up-regulated HSP47 expression.

We have also now demonstrated the knockdown of HSP47 in primary HSCs after treatment with LNPs as the reviewer suggested (Fig. 4E). The results showed that AA-T3A-C12/siHSP47 LNP achieved ~65% knockdown of HSP47, which is much higher than MC3/siHSP47 LNP. The corresponding discussion has now been added to the manuscript (Page 9, line 264) as follows: “Afterwards, their silencing activity was evaluated in activated primary HSCs, which were confirmed to express a much higher level of sigma receptors than hepatocytes (Fig. S18). The results showed that AA-T3A-C12/siHSP47 LNP achieved significantly higher knockdown efficiency compared to MC3/siHSP47 LNP (65% vs 34%, Fig. 4E).”

Fig. 4. AA-T3A-C12 LNP-mediated GFP and HSP47 knockdown in activated fibroblasts. (D,E) Western blot analysis of HSP47 expression in LNP-treated activated 3T3 fibroblasts (D) and primary HSCs (E).

3) Figure 5A: I wonder if the signals that are shown are really signals from the LNP. An untreated control is missing. I know from my own experiences that using the IVIS imaging device, also untreated livers appear red - due to the autofluorescence of the liver. Thus, an untreated group of mice must be added here. Assure that you use the same IVIS settings again.

Response 3: We thank the reviewer for this important suggestion. The IVIS imaging results of PBS-treated control mice are now added to Fig 5A with the same IVIS settings. There were very weak signals in the liver and other organs of control mice.

Fig. 5. Biodistribution and HSP47 silencing activity of LNPs in fibrotic mice. (A) *Ex vivo* fluorescence imaging and signal quantification of major organs from **PBS**, AA-T3A-C12 LNP/Cy5-siRNA or MC3 LNP/Cy5-siRNA treated fibrotic mice (n = 3).

4) A comparison of their formulation to the one from Sato et al 2008, *Nature Biotechnology*, in which they used Vitamin A-loaded nanoparticles would, to my mind, be a better comparison than towards MC3-LNP. MC3 is optimized for a release in hepatocytes and not for stellate cells. Therefore, it can not be a general benchmark for knockdown, only for a knockdown in hepatocytes.

Response 4: We thank the reviewer for this important suggestion. Currently, there are no commercially available *in vivo* transfection reagents for efficient gene delivery to HSCs. Due to the critical role of this cell population in liver fibrosis and the lack of efficient delivery vehicles to target these cells, we have engineered an ionizable lipid with a targeting ligand in order to enhance siRNA delivery to HSCs. MC3 is a benchmark ionizable lipid, which has been shown to transfect not only hepatocytes but also other cell types including HSCs (*Mol. Pharmaceutics* 2018, 15, 1, 53-62) and immune cells (*Nano Lett.* 2022, 22, 8304-8311). In order to address the HSC-targeting ability of our AA-T3A-C12 LNP, we utilized MC3 LNP with similar organ distribution patterns for comparison (**Fig. 5A**).

As the reviewer suggested, we tried to include the previously reported vitamin A-loaded liposome (i.e., VA-lip-siHSP47) for comparison (**Fig. 1R4**, see below). VA-lip-siRNA was formulated using vitamin A, DC-6-14, cholesterol, DOPE at a molar ratio of 8:4:3:3 according to the publication (*Nat. Biotechnol.* 2008, 26, 431-442). However, we found that the resulting VA-lip-siRNA was cloudy and its size and PDI were much larger than our AA-T3A-C12/siRNA LNP or MC3/siRNA LNP (**Fig. 1R4A**), most likely due to the lack of a stabilizing component (i.e. PEGylated lipid) in this formulation. Moreover, we found that cell viability was less than 80% when cells were treated with VA-lip-siGFP at an siRNA dose of 50 nM (**Fig. 1R4B**), which was not observed with our AA-T3A-C12/siGFP LNP (**Fig. S15**). Since DC-6-14 is a permanently positive-charged lipid and the resulting liposome is positively charged (25.7 mV), it is expected to be more toxic than our ionizable lipid-based neutral LNPs. We further compared the knockdown efficiency of VA-lip-siHSP47 with our AA-T3A-C12/siHSP47 LNP in primary HSCs (**Fig. 1R4C**). The results showed that VA-lip-siHSP47 was much less potent than our

AA-T3A-C12/siHSP47 LNP in silencing HSP47. Additionally, we found that VA-lip-siRNA accumulated poorly in the liver compared to AA-T3A-C12 LNP (Fig. 1R4D and Fig. 5A). A strong Cy5 signal was observed in the kidneys, presumably due to the leakage of Cy5-siRNA from this unstable liposome. Additionally, a strong signal was observed in the lungs, which is typically observed for cationic liposomes (*Journal of controlled release*, 2000, 69(1), 139-148). Due to the dramatic differences with regards to physio-chemical properties, cytotoxicity and biodistribution between VA-lip-siHSP47 and our LNP, we decided not to choose VA-lip-siHSP47 for comparison and not to include the data below in the manuscript.

Fig. 1R4. Characterization and performance of VA-lip-siRNA. (A) Physio-chemical parameters of LNPs. A photograph of LNPs in 1.5 mL Eppendorf tubes is shown. (B) Cell viability after treatment with VA-lip-siGFP for 48 h. 3T3 cells were treated with indicated siRNA doses for 48 h. (C) Western blot analysis of HSP47 expression in LNP-treated activated primary HSCs. HSCs were treated with LNPs at an siRNA dose of 50 nM for 48 h. (D) *Ex vivo* fluorescence imaging and signal quantification of major organs from VA-lip-siRNA treated mice. Cy5-siRNA (5 μg/mouse) was delivered by this liposome formulation into fibrotic mice and *ex vivo* imaging was performed 1 h post-injection. Data are presented as mean ± SD (n = 3). ****p* <

0.001.

Reviewer #2 (Remarks to the Author):

This is a well written manuscript. The authors have generated a novel small molecule tethered LNP that delivers really potent siRNA based system as compared to the current gold standard. The authors elegantly show silencing and functional knock down in liver hepatic stellate cells. Targeting HSCs has remained a challenge and this work shows it is possible to target this cell population. I recommend publication with minor revisions.

Response: We very much appreciate the reviewer's encouraging comments, as well as their time and effort in helping us improve the manuscript.

1. Is there an effect of anisamide as a small molecule in fibrosis. Will treatment in even cell types show that anisamide alone does not affect the fibrotic events. One experiment that uses anisamide in MC3 formulations or separately to show that effects are due to silencing and not an effect of anisamide in the formulation might improve the impact.

Response 1: We thank the reviewer for this suggestion. The small molecule anisamide has never been reported to affect fibrotic progress or HSP47 expression. According to our results, AA-T3A-C12/siGFP LNP does not affect HSP47 expression (**Fig. 4D**), suggesting that AA-T3A-C12/siHSP47 LNP-mediated down-regulation of HSP47 was not due to anisamide.

However, to directly address the reviewer's concern, we have now performed a Western blot assay to check the HSP47 silencing activity of MC3 LNP with or without free anisamide (i.e., 4-methoxybenzoic acid) treatment. The results showed that 4-methoxybenzoic acid treatment did not alter the HSP47 expression in MC3/siGFP LNP-treated cells (**Fig. 2R1**, see below). However, using siHSP47 instead of siGFP in MC3 LNP greatly reduced HSP47 expression. These results confirmed that the down-regulation of HSP47 was due to the silencing effect of siHSP47, and not the effect of anisamide. Since MC3 LNP is not the focus of this study, and it is clear that AA-T3A-C12/siGFP LNP does not alter the expression of HSP47 (**Fig. 4D**), we decided not to include the data below in the manuscript.

Fig. 2R1. Western blot analysis of HSP47 expression. Activated 3T3 fibroblasts were treated with MC3/siGFP LNP or MC3/siHSP47 LNP (50 nM siRNA) with or without 4-methoxybenzoic acid (10 μ M) for 48 h. GAPDH was used as an internal control. Data are presented as mean \pm SD (n = 3). ns, not significant; *** p < 0.001.

2. Is there a dose dependence in silencing. IC50 for silencing might be helpful or is the max silencing that can be achieved is limited to 65%.

Response 2: We thank the reviewer for this helpful suggestion. We conducted a therapeutic study involving four i.v. injections of 5 μ g siHSP47 per mouse, and achieved 65% knockdown at the end of the experiment in the AA-T3A-C12/siHSP47 LNP group (**Fig. 5B-F**). IC50, or more specifically ED50 (median effective dose), is usually calculated based on the result of a single treatment. In order to obtain ED50 for AA-T3A-C12/siHSP47 LNP, we have now tested its silencing activity at different doses in fibrotic mice after a single dosage (**Fig. S21**). Western blot analysis of hepatic HSP47 expression showed that there was a dose-dependent silencing activity of AA-T3A-C12/siHSP47 LNP and ~70% knockdown efficiency can be achieved at a dose of 0.5 mg/kg (12.5 μ g siHSP47/mouse). The ED50 was determined to be 0.34 mg/kg. The corresponding discussion has been added to the manuscript (Page 11, line 311) as follows: “The siRNA dose was pre-determined to be 5 μ g (0.2 mg/kg) for each injection, which was lower than the median effective dose (ED50 = 0.34 mg/kg) for AA-T3A-C12/siHSP47 LNP but still achieved moderate HSP47 knockdown (**Fig. S21**).”

Figure S21. Western blot analysis of HSP47 expression in the liver. Mice were i.p. injected with 20% CCl₄ (0.7 μ l/g) in corn oil twice a week for 2 weeks and then i.v. injected with AA-T3A-C12/siHSP47 LNP at an siRNA dose of 5 μ g/mouse (0.2 mg/kg) or 12.5 μ g/mouse (0.5 mg/kg). Mice were euthanized 3 days after treatment and livers were harvested for western blot analysis. GAPDH was used as an internal control. Data are presented as mean \pm SD (n = 3).

3. Fig 5 legends is unclear with the numerical numbers.

Response 3: We thank the reviewer for pointing this out. We have now revised this part and have made the legend clear. The numerical numbers were replaced by G1-G5. Detailed information is now provided in both **Fig. 5** legends as follows “G1, healthy mice; G2, PBS-treated fibrotic mice; G3, AA-T3A-C12/siGFP LNP-treated fibrotic

mice; G4, AA-T3A-C12/siHSP47 LNP-treated fibrotic mice; G5, MC3/siHSP47 LNP-treated fibrotic mice.” We have now also updated in the manuscript (Page 11, line 308) as follows “Healthy mice were used as an untreated control (Group 1, G1). During the last two weeks of the experiment, fibrotic mice were treated twice weekly with PBS (G2), AA-T3A-C12/siGFP LNP (G3), AA-T3A-C12/siHSP47 LNP (G4) or MC3/siHSP47 LNP (G5).”

4. Overall, this is a well written work.

Response 4: We thank the reviewer for this encouraging comment, and hope that they enjoy our revised manuscript!

Reviewer #3 (Remarks to the Author):

The manuscript entitled **Ligand-Tethered Lipidoids for Targeted RNA Delivery to Treat Liver Fibrosis** by Han et. al developed a combinatorial library of anisamide ligand-tethered lipidoids (AA-lipidoids) to identify AA-lipidoids with both high potency and selectivity to deliver RNA payloads to activated fibroblasts. In a preclinical model of liver fibrosis, AA-lipidoids reduced collagen deposition and liver fibrosis. Most experimental evidence is supported with proper experimental data. But still, the manuscript lacks in few portions which the authors need to address before its acceptance for publication in a high-quality journal like Nature Communications. Below are the Major comments that the authors have to address to improve the overall quality of the manuscript.

Response: We very much appreciate the reviewer's helpful comments, as well as their time and effort in helping us improve the manuscript.

1. Detailed information on the microfluidic chip was not mentioned in this manuscript. To improve the transparency of the work, the design of microfluidic chips should be provided. Main experimental parameters, such as the flow rate, are needed to be described.

Response 1: We thank the reviewer for these important suggestions. The design of microfluidic chips was reported in a previous publication (*J. Am. Chem. Soc.* 2012, 134, 6948-6951). We have now added this information to the **LNP formulation** section (Page 15, line 497) as follows: "The aqueous phase and organic phase were mixed at an ionizable lipid:RNA weight ratio of 10:1 and at a flow rate of 1.8 mL/min and 0.6 mL/min (3:1) using Pump 33 DDS syringe pumps (Harvard Apparatus, MA) in a microfluidic device with a staggered herringbone micromixer design. The microfluidic devices were fabricated in polydimethylsiloxane according to standard soft lithographic procedures³⁷. A two-step exposure process was used to create the SU-8 master with positive channel features on a silicon wafer. Each mixing channel is 4 cm in length. LNPs were dialyzed against 1× PBS in a 20kDa MWCO cassette for 2 h, filtered through a 0.22 μM filter, and stored at 4 °C."

2. How is the optimal ratio of nucleic acid to LNP determined? This important parameter must be identified by agarose gel electrophoresis (Nano Letters, 2018, 18(6): 3328-3334.).

Response 2: We thank the reviewer for bringing up this question. The weight ratio of ionizable lipid to nucleic acid (10:1) was optimized in a previous publication (*Nano Lett.* 2015, 15, 7300-7306) and widely adopted by others and us (*Nat. Biotech.*, 2019, 37(10), 1174-1185; *RSC Adv.*, 2022,12, 25397-25404), which typically results in high encapsulation efficiency (> 85%) of nucleic acids. However, to address the reviewer's

concern, we have now performed both agarose gel electrophoresis and a RiboGreen RNA assay to confirm the encapsulation of siRNA in AA-T3A-C12 LNP and these results have now been added into **Fig. S8**. The results showed that when the weight ratio of AA-T3A-C12:siRNA was 10:1, approximately no free siRNA was present (**Fig. S8A**) and the siRNA encapsulation efficiency reached a plateau of ~90% (**Fig. S8B**). The corresponding materials, methods, and discussion were updated in the manuscript as follows:

Materials (Page 14, line 466) - “1% agarose gels with SYBR™ Safe (#A42100) were purchased from ThermoFisher.”

Methods (Page 15, line 511) - “The siRNA-LNP complex was analyzed by agarose gel electrophoresis⁴⁸.”

Discussion (Page 6, line 177) - “Afterwards, the four-component AA-T3A-C12/siRNA LNP was formulated by microfluidic mixing at an ionizable lipid:siRNA weight ratio of 10:1³⁷⁻³⁹. This weight ratio was chosen based on the results of both the gel retardation assay and the RiboGreen RNA assay, which resulted in high siRNA encapsulation efficiency (EE = $87.4 \pm 3.8\%$, **Fig. S8** and **Table S1**).”

The publication (*Nano Letters*, 2018, 18(6): 3328-3334) is now cited as reference 48.

Figure S8. Optimization of the weight ratio of AA-T3A-C12 to siRNA. (A) Gel retardation assay. LNPs with different weight ratios of AA-T3A-C12:siRNA were analyzed by 1% agarose gel electrophoresis. 500 ng siRNA was loaded into each lane. Barely no free siRNA was observed when the weight ratio of AA-T3A-C12:siRNA was above 5:1. (B) RiboGreen RNA assay. The siRNA encapsulation efficiency reached a plateau of ~90% when the weight ratio of AA-T3A-C12:siRNA was 10:1. Data are presented as mean \pm SD (n = 3).

3. The characterization of the nanoparticles is insufficient. For example, TEM images of LNP vector without DNA encapsulation should be added. Stability tests of LNPs in protein, as well as in serum, were completely missing.

Response 3: We thank the reviewer for these important comments. More characterization of LNPs is now provided. A representative TEM image of empty AA-T3A-C12 LNP has now been added to **Fig. S9**. More physical parameters of empty AA-T3A-C12 LNP are now summarized in **Table S1**. The stability of LNPs in protein and in serum was studied and the results are now presented in **Fig. S10**. The corresponding discussion has now been updated in the manuscript (Page 6, line 181) as follows: “The hydrodynamic diameter of these LNPs was approximately 65.6 nm with a narrow polydispersity index (PDI) of 0.018, which was slightly larger than empty LNPs (**Table S1**). Moreover, these LNPs had a neutral surface charge and a pKa of 5.72. Transmission electron microscopy (TEM) images showed that both empty LNPs and siRNA-loaded LNPs had a spherical morphology (**Fig. 3A** and **Fig. S9**), with some collapsed LNPs due to the dehydration process of sample preparation. Additionally, these LNPs had a high colloidal stability, as large LNP aggregates (> 200 nm) were not observed in the protein- or serum-supplemented environment at 37 °C for 48 h (**Fig. S10**). Overall, the above results suggest the formation of homogeneous and stable LNPs with high siRNA EE that are promising for potent delivery *in vitro* and *in vivo* due to their small size (< 100 nm), neutral charge, and suitable pKa (5.5–7.0)⁴⁰.”

Figure S9. A representative TEM image of empty AA-T3A-C12 LNP. Scale bar, 100 nm.

Table S1. Characterization of LNPs.

LNP	Diameter (nm)	PDI	Zeta potential (mV)	Encapsulation efficiency (%)	pKa
Empty AA-T3A-C12 LNP	59.3 ± 2.5	0.021	-0.7 ± 0.9	n/a	n/a
AA-T3A-C12/siRNA LNP	65.6 ± 1.2	0.018	-1.1 ± 1.5	87.4 ± 3.8	5.72
T3A-C12/siRNA LNP	64.9 ± 1.8	0.034	-2.2 ± 1.8	33.6 ± 8.5	5.81
MC3/siRNA LNP	60.9 ± 3.4	0.090	-1.5 ± 2.2	90.1 ± 2.5	6.44 ^a

^aThis data was obtained from the following publication: Jayaraman, M. *et al.* Maximizing the potency of siRNA lipid nanoparticles for hepatic gene silencing in vivo. *Angewandte Chemie* **51**, 8529-8533 (2012).

Figure S10. Hydrodynamic sizes of AA-T3A-C12/siRNA LNP. 1% bovine serum albumin (BSA) and 10% FBS were prepared in 1×PBS and filtered through 0.22 μm filters to remove large protein aggregates. LNPs were then diluted in 1×PBS, 1% BSA/PBS or 10% FBS/PBS and incubated at the indicated temperature for 48 h. The average hydrodynamic size was determined by DLS. The absorption of protein on LNPs increased the size, but large LNP aggregates (> 200 nm) were not observed. Data are presented as mean ± SD (n = 3).

4. The difficulty of gene transfection in primary cells (e.g. 3T3 cells and H2.35 cells) is significantly higher than that in cell lines. The efficacy of LNP should be tested in primary hepatocytes.

Response 4: We thank the reviewer for this helpful suggestion. Since activated HSCs are identified as the primary source of HSP47 in the fibrotic liver (Lab. Invest. 2005, 85, 789-797), we have now further examined the knockdown efficiency of our LNPs in primary HSCs (**Fig. 4E**). The results showed that AA-T3A-C12/siHSP47 LNP achieved ~65% knockdown of HSP47, which is much higher than MC3/siHSP47 LNP. The corresponding discussion has now been added to the manuscript (Page 9, line 264) as follows: “Afterwards, their silencing activity was evaluated in activated primary HSCs, which were confirmed to express a much higher level of sigma receptors than hepatocytes (**Fig. S18**). The results showed that AA-T3A-C12/siHSP47 LNP achieved significantly higher knockdown efficiency compared to MC3/siHSP47 LNP (65% vs 34%, **Fig. 4E**).”

Fig. 4. AA-T3A-C12 LNP-mediated GFP and HSP47 knockdown in activated fibroblasts. (D,E) Western blot analysis of HSP47 expression in LNP-treated activated 3T3 fibroblasts (D) and primary HSCs (E).

5. The efficacy of LNP was not comprehensive as evidenced by body weight data and immunofluorescence staining alone. More intuitive and convincing experiments such as ICG clearance tests, blood AST, blood ALT, and blood TBIL should be provided.

Response 5: We thank the reviewer for this important suggestion. We previously performed AST and ALT tests, and these results were in the previous **Fig. 6C,D**. As suggested by the reviewer, we have now performed a TBIL test using the preserved serum, and the result has now been added into **Fig. 6E**. The corresponding discussion has now been added into the manuscript (Page 13, line 373) as follows: “**First, serum alanine aminotransferase (ALT), aspartate aminotransferase (AST) and total bilirubin (TBIL) were quantified to assess liver toxicity (Fig. 6C-E). ALT, AST and TBIL markedly increased in fibrotic mice compared to healthy mice, as a result of CCl₄-induced liver injury. AA-T3A-C12/siGFP LNP treatment did not further increase ALT, AST and TBIL levels, indicating that AA-T3A-C12 LNP was well-tolerated without exacerbating liver injury. Moreover, AA-T3A-C12/siHSP47 LNP treatment slightly decreased ALT, AST and TBIL levels in fibrotic mice, but these decreases were not significant.**”

Fig. 6. Macroscopic, histopathological and biochemical analysis of liver fibrosis. (C) Quantification of serum alanine aminotransferase (ALT). (D) Quantification of serum aspartate aminotransferase (AST). (E) Quantification of serum total bilirubin (TBIL). Data are presented as mean \pm SD (n = 5). ns, not significant; * p < 0.05; *** p < 0.001.

6. The contraction and death of hepatocytes are important indicators of liver fibrosis. Images of H&E sections in Fig. 6 should be enlarged to observe changes in cell morphology.

Response 6: We thank the reviewer for this helpful suggestion. We have now enlarged the pictures and updated **Fig. 6A**. The corresponding discussion has now been updated in the manuscript (Page 12, line 348) as follows: “Additionally, extensive contraction and death of hepatocytes was observed in CCl₄-treated mice. However, livers from AA-T3A-C12/siHSP47 LNP-treated fibrotic mice exhibited a more normal histological structure and had less fibrotic septa, as well as less apoptotic hepatocytes compared to other treatment groups.”

Fig. 6. Macroscopic, histopathological and biochemical analysis of liver fibrosis. (A) Representative images of livers and liver sections stained with hematoxylin and eosin (H&E) or Picosirius red. Black arrows indicate apoptotic hepatocytes.

7. The use of symbols in manuscripts is very casual. For example, throughout the manuscript, the multiplication sign is written as the letter x. The authors should carefully check the paper.

Response 7: We thank the reviewer for pointing out this problem. We have now corrected this throughout the manuscript.

8. Authors claimed that LNP was subsequently uptake by perivascular HSCs (Page 10, line 13). However, immunofluorescence did not differentiate between LNP phagocytosis by HSCs and hepatic macrophages. More evidence is needed to prove this claim.

More evidence is needed to prove the claim.

Response 8: We thank the reviewer for raising this important question. To verify the HSC-targeting ability of LNPs, we have now further stained for α -smooth muscle actin (α -SMA, a marker of activated HSCs). The results showed that more AA-T3A-C12 LNP co-localized with or were close to activated HSCs compared to MC3 LNP (Fig. S20). These results demonstrate the superiority of AA-T3A-C12 LNP to target activated HSCs. The corresponding methods have now been added into the manuscript (Page 17, line 598) as follows: “Livers were collected to prepare 10 μ m cryosections. Samples were stained with FITC-conjugated α -smooth muscle actin antibody (#F3777, Sigma) at a 1:500 dilution overnight at 4 °C. Nuclei were stained with Hoechst 33342 (10 μ g/mL) before images were taken using a confocal laser scanning microscope.” The corresponding discussion has now been updated in the manuscript (Page 9, line 281) as follows: “To further investigate the HSC-targeting ability of LNPs, livers were cryosectioned and α -smooth muscle actin (α -SMA, a marker of activated HSCs) was stained for immunofluorescence analysis. Confocal imaging results showed that more AA-T3A-C12 LNP co-localized with or were close to activated HSCs compared to MC3 LNP (Fig. S20), presumably due to the strong affinity between anisamide ligands and sigma receptors. Together, these results demonstrate the superiority of AA-T3A-C12 LNP to target the liver and activated HSCs.”

Figure S20. Representative confocal images of liver sections. Fibrotic mice were i.v. injected with AA-T3A-C12 LNP/Cy5-siRNA or MC3 LNP/Cy5-siRNA at a dose of 5 μ g siRNA/mouse. 1 h post-injection, mice were euthanized and livers were collected for immunofluorescence staining. Compared to MC3 LNP, more AA-T3A-C12 LNP co-localized with or were close to HSCs (α -SMA⁺). Arrows indicate central veins of liver lobules. Scale bar: 100 μ m.

Reviewer #4 (Remarks to the Author):

Remarks to the authors:

Han and his team have done a good job in developing a ligand-targeted LNP-siRNA delivery system for a potential liver fibrosis gene therapy. In their research, an efficient 2-step in vitro LNP screening method was also established to contribute to the potential fast-track development of clinical and pharmaceutical applications. The highlight of their research is to report the noteworthy efficacy of the candidate AA-T3A-C12 LNP in the knockdown of the disease-related overexpressing HSP47 protein in liver fibroblasts. The manuscript was well-written with a thorough literature review citing the most important recent papers in LNP and LNP-targeting delivery. A significant amount of work was conducted, including synthesis, in vitro screening and in vivo biodistribution, efficacy and safety studies. Overall, the research is thoughtful and insightful, although some experiments need to be adjusted and added.

Response: We very much appreciate the reviewer's encouraging comments, as well as their time and effort in helping us improve the manuscript.

Major comments:

1. For the in vitro screening part, it was mentioned in the discussion that the 3T3 cell line was chosen because it expresses the sigma receptors. However, no evidence showed that the sigma reporters are also overexpressed in primary HSCs. Therefore, it might not show the relevance of settling the dependency on sigma receptor-mediated transfection efficiency as the round 2 selection, unless it was meant to be the indicator for 3T3/hepatic cells selectivity of the LNP expression only. Although the 3T3/hepatic cells co-culture results were impressive, it still could not support the conclusion that the AA-T3A-C12 LNP would be the best formulation in vivo based on the in vitro studies if it was settled like this. The statement of "both passive and active targeting are important criteria" for LNP delivery could not be made if this has not been confirmed in primary liver fibroblasts. Addition of the primary cell line data of liver fibroblasts, or replacing the 3T3 cell line with primary liver fibroblasts showing the expression of sigma receptors (e.g. western blot), would be a more suitable model for this experiment.

Response 1: We thank the reviewer for these helpful suggestions. Previous studies have confirmed the increased expression of sigma receptors in activated murine 3T3 cells and an activated human HSC cell line (LX-2) using western blot (*ACS Nano*, 2016, 10, 9243-9258; *Nat. Nanotechnol.*, 2021, 16, 466-477). Moreover, sigma receptors were confirmed to be significantly up-regulated in activated HSCs in metastatic cancer mouse models as well as in fibrotic livers from patients using immunofluorescence staining (*Nat. Commun.*, 2019, 10, 2993; *Nat. Nanotechnol.*,

2021, 16, 466-477).

To address the reviewer's concern, we have now conducted western blots to analyze the expression of sigma receptors in hepatocytes, primary HSCs and activated primary HSCs (**Fig. S18**). The results demonstrate that sigma receptors are expressed in primary HSCs and are further up-regulated in activated primary HSCs, similar to the previously reported 3T3 cells (*ACS Nano*, 2016, 10, 9243-9258). However, hepatocytes had negligible sigma receptor expression. We agree with the reviewer that the primary HSCs/hepatocytes co-culture would be a more suitable model instead of 3T3 cells/hepatocytes, but the lack of reporter genes (e.g., GFP) for both primary HSCs and H2.35 hepatocytes makes us unable to perform the flow cytometry analysis similar to **Fig. 3C**.

Figure S18. Western blot analysis of sigma receptor expression in immortalized primary hepatocytes H2.35, primary HSCs and activated primary HSCs. Activated HSCs were obtained by stimulation with 10 ng/ml of TGF- β for 24 h. GAPDH was used as an internal control. The expression of sigma receptor was normalized to hepatocytes. The expression of sigma receptor was much higher in HSCs than hepatocytes. Moreover, activated HSCs further up-regulated sigma receptors.

To address the reviewer's concern, we examined the knockdown efficiency of our LNPs in primary HSCs (**Fig. 4E**). The results showed that AA-T3A-C12/siHSP47 LNP achieved ~65% knockdown of HSP47, which is much higher than MC3/siHSP47 LNP. The corresponding discussion has now been added to the manuscript (Page 9, line 264) as follows: "Afterwards, their silencing activity was evaluated in activated primary HSCs, which were confirmed to express a much higher level of sigma receptors than hepatocytes (**Fig. S18**). The results showed that AA-T3A-C12/siHSP47 LNP achieved significantly higher knockdown efficiency compared to MC3/siHSP47 LNP (65% vs 34%, **Fig. 4E**)."

Fig. 4. AA-T3A-C12 LNP-mediated GFP and HSP47 knockdown in activated fibroblasts. (D,E) Western blot analysis of HSP47 expression in LNP-treated activated 3T3 fibroblasts (D) and primary HSCs (E).

To further support our claim, we have now further verified the HSC-targeting ability of AA-T3A-C12 LNP *in vivo* by immunofluorescence staining of α -smooth muscle actin (α -SMA, a marker of activated HSCs). The results showed that more AA-T3A-C12 LNP co-localized with or were close to activated HSCs compared to MC3 LNP (Fig. S20), which suggests the superiority of AA-T3A-C12 LNP to target activated HSCs. The corresponding discussion has now been updated in the manuscript (Page 9, line 281) as follows: “To further investigate the HSC-targeting ability of LNPs, livers were cryosectioned and α -smooth muscle actin (α -SMA, a marker of activated HSCs) was stained for immunofluorescence analysis. Confocal imaging results showed that more AA-T3A-C12 LNP co-localized with or were close to activated HSCs compared to MC3 LNP (Fig. S20), presumably due to the strong affinity between anisamide ligands and sigma receptors. Together, these results demonstrate the superiority of AA-T3A-C12 LNP to target the liver and activated HSCs.”

Figure S20. Representative confocal images of liver sections. Fibrotic mice were i.v. injected with AA-T3A-C12 LNP/Cy5-siRNA or MC3 LNP/Cy5-siRNA at a dose of 5 μg siRNA/mouse. 1 h post-injection, mice were euthanized and livers were collected for immunofluorescence staining. Compared to MC3 LNP, more AA-T3A-C12 LNP co-localized with or were close to HSCs ($\alpha\text{-SMA}^+$). Arrows indicate central veins of liver lobules. Scale bar: 100 μm .

2. Similarly, it appears that the feature of high dependency on sigma receptor-mediated transfection efficiency was chosen as the key and the only indication for fibroblast/hepatocyte selectivity in the comparison of AA-T3A-C12 and MC3 LNPs. However, no data was shown as to whether this feature may also impact the cellular uptake of MC3 LNP. This might not be a fair comparison to compare the ligand-targeted nanoparticle (AA-T3A-C12 LNP) with the particles following passive diffusion (MC3 LNP). More investigation on AA ligand-targeted MC3 LNP would be required if it is meant to prove that the reported lipid is more efficient than MC3. Also, it would make more sense if the MC3 LNP was also shown in the 2-step screening process as a control. It could be much easier to see the differences between AA-T3A-C12 and MC3 LNPs if the left Y-axis (MFI of siRNA) can be modified (e.g. % or 0 to 1 measurement/control) in some figures.

Response 2: We thank the reviewer for these important comments. We have developed a one-pot, two-step synthetic method to modify polyamines and synthesize ligand-tethered lipidoids. However, we are unable to modify the structure of the MC3 lipid, which is approved by the FDA as a key component in Onpattro[®]. MC3 has been widely used as a benchmark ionizable lipid, and many studies have compared their in-house ionizable lipids with MC3 to highlight their strong potency (*Sci. Adv.*, 2021, 7, eaba1028; *PNAS*, 2021, 118, 10, e2020401118). In this study, we demonstrated that

our lead lipid AA-T3A-C12 mediated greater RNA delivery and transfection in activated fibroblasts/HSCs than MC3 (Figs. 3C, 4D and 4E).

As the reviewer suggested, we have now tested the knockdown efficiency of siGFP-loaded MC3 LNP with or without HP pre-treatment in activated 3T3-GFP fibroblasts. The results have now been added into Fig. S5 and the corresponding discussion has been now updated in the manuscript (Page 6, line 163) as follows: “For lipidoids without anisamide, no obvious loss of silencing activity was observed after HP treatment ($p = 0.508$, Fig. 2D, E). Similar results were observed when the FDA-approved MC3 LNP formulation was tested (Table S1 and Fig. S5).” Since the GFP knockdown efficiency of MC3 LNP was less than 80% and it did not meet the criteria to enter the second round of screening, we did not put these results into Fig. 2B, D.

Figure S5. MC3 LNP-mediated GFP knockdown. Activated 3T3-GFP fibroblasts were pre-treated with or without 30 μ M haloperidol (HP) for 2 h before GFP siRNA-loaded MC3 LNP (50 nM) was used to treat cells for another 48 h. No obvious loss of silencing activity was observed after HP treatment. Data are presented as mean \pm SD ($n = 3$). ns, not significant.

We are grateful for the reviewer’s suggestion about “the left Y-axis (MFI of siRNA) can be modified (e.g. % or 0 to 1 measurement/control) to make it easier to see the differences between AA-T3A-C12 and MC3 LNPs”. However, we prefer to use the absolute values instead of the relative ones, as they show the more objective and straightforward results. We agree with the reviewer that it will make it easier to see the fold changes. Therefore, we have now improved Fig. 3C and Fig. S13 by marking the fold changes.

Fig. 3. Characterization of AA-T3A-C12 LNP and cellular uptake. (C) Flow cytometry analysis of competitive cellular uptake of Cy5-siRNA-loaded LNP in a fibroblast/hepatocyte (3T3-GFP/H2.35) co-culture environment. The mean fluorescence intensity ratio between fibroblast and hepatocyte ($MFI_{3T3-GFP}/MFI_{H2.35}$) was calculated to indicate preferential uptake by fibroblasts over hepatocytes. Data are presented as mean \pm SD ($n = 3$). $**p < 0.01$; $***p < 0.001$.

Figure S13. Flow cytometry analysis of competitive cellular uptake of Cy5-siRNA-loaded AA-T3A-C12 LNP or T3A-C12 LNP in a fibroblast/hepatocyte (3T3-GFP/H2.35) co-culture environment. The mean fluorescence intensity ratio between fibroblast and hepatocyte ($MFI_{3T3-GFP}/MFI_{H2.35}$) was calculated to indicate the preferential uptake of LNP by fibroblasts over hepatocytes. The $MFI_{3T3-GFP}/MFI_{H2.35}$ of AA-T3A-C12 LNP was 0.56, which was significantly higher than 0.36 of T3A-C12 LNP. Data are presented as mean \pm SD ($n = 3$). $**p < 0.01$; $***p < 0.001$.

Minor comments:

3. In the Introduction, it was said that “However, to the best of our knowledge, lipidoids that intrinsically mediate efficient RNA delivery into activated HSCs have not been reported”. Actually, there are few papers that have been published on this topic, such as Specific hepatic delivery of procollagen $\alpha 1(I)$ siRNA in lipid-like nanoparticles resolves liver fibrosis (doi: 10.1002/hep.27936) and Comparison of DLin-MC3-DMA and ALC-0315 for siRNA Delivery to Hepatocytes and Hepatic Stellate Cells (doi:

<https://doi.org/10.1021/acs.molpharmaceut.2c00033>). It would be beneficial to add these papers to the references, especially the second one to support the comparison with MC3 LNPs reported in the manuscript.

Response 3: We thank the reviewer for this important suggestion. We have now added these two papers to the references and have now revised this sentence (Page 3, line 71) as follows: “To the best of our knowledge, lipidoids that intrinsically mediate targeted RNA delivery to activated HSCs have not been reported, although some LNPs have been demonstrated to passively target these cells^{27, 28}.”

4. The authors should clarify if the formulation of the MC3 LNP that has been used for comparison is the standard formulation (MC3/DSPC/cholesterol/DMG-PEG= 50/10/38.5/1.5, N/P 6). It is noticeable that the reported lipid has a giant headgroup, which might lead to some issues of low efficiency of siRNA loading. This could be the reason for the slightly adjusted LNP formulation and the N/P ratio 10 of the AA-T3A-C12 LNP. It may lead to unfair comparison without notification of the formulation MC3 LNP, as the formulation can make considerable differences in transfection efficiency, gene expression levels and toxicity.

Response 4: We thank the reviewer for these helpful comments. We have corrected some information and provided more details in the **LNP formulation** section (Page 15, line 494) as follows: “An organic phase was prepared in ethanol by solubilizing ionizable lipid (lipidoid, AA-lipidoid or MC3), DSPC, cholesterol and C14-PEG2000 at a molar ratio of 50:10:38.5:1.5. The aqueous phase was prepared in 10 mM citrate buffer (pH 3) with siRNA or luciferase mRNA. The aqueous phase and organic phase were mixed at an ionizable lipid:RNA weight ratio of 10:1 and at a flow rate of 1.8 mL/min and 0.6 mL/min (3:1) using Pump 33 DDS syringe pumps (Harvard Apparatus, MA) in a microfluidic device with a staggered herringbone micromixer design.”

We fixed the weight ratio of ionizable lipid:nucleic acid at 10:1 throughout the study. This weight ratio was optimized in a previous publication (*Nano Lett.* 2015, 15, 7300-7306) and widely adopted by others and us (*Nat. Biotech.*, 2019, 37(10), 1174-1185; *RSC Adv.*, 2022,12, 25397-25404), which typically results in high encapsulation efficiency (> 85%) of nucleic acids. In this case, N/P ratio is not fixed and can vary based on the molecular weight of ionizable lipid and the number of amines in each lipid. For MC3 LNP, the N/P ratio is calculated to be 5.2, which is close to the standard MC3 LNP mentioned by the reviewer. For AA-T3A-LNP, the N/P ratio is calculated to be 9.

5. The LNP manufacturing method may raise concerns about the pH values of the LNP samples after 2 hours of dialysis. In previous research, it mentioned that 4- 6 hours of dialysis was essential to be performed to alter the pH of the samples

from 4 to 7.4 instead of 2 hours from pH 3 to 7.4 in this research. The pH of LNP samples may also impact the in vitro and in vivo results. Did the authors measure the pH of the LNP samples after dialysis?

Response 5: We thank the reviewer for raising this concern. We measured the pH after 2 hours of dialysis to be around 6.9. Although the pH does not reach 7.4 after 2 hours of dialysis, we would further dilute these LNPs in 1×PBS or culture medium before animal or cell treatment, in which the pH was measured to be around 7.4. We agree with the reviewer that extension of the dialysis period can bring the pH close to 7.4, however, we dialyze for 2 hours by taking time cost, risks of contamination, and nucleic acid degradation into consideration.

REVIEWERS' COMMENTS

Reviewer #2 (Remarks to the Author):

I recommend publication

Reviewer #3 (Remarks to the Author):

In the revised manuscript, the authors have addressed all my previous comments. Therefore, I recommend acceptance of this manuscript without the need for further revisions.

Reviewer #4 (Remarks to the Author):

Remarks to the authors:

Thank you so much for sending this well-structured and well responded rebuttal letter back to us. I think all my questions and concerns have been answered in a decent way. I would like to recommend and to support this remarkable manuscript to be published in Nature Communications.

Han and his team has done a superb job. They not only presented massive amount of the noteworthy data, also inspire other researchers in this and related fields. This high quality paper can be a model for research in development of the ligand-targeting nucleic acid delivery systems and potentially pre-clinical therapies. Overall, this is a well-written manuscript, great job!

REVIEWERS' COMMENTS AND AUTHORS' ANSWERS

Note: Our responses (standard typeface) to reviewers' comments (bold).

Reviewer #2 (Remarks to the Author):

I recommend publication.

Response: We thank the reviewer again for their helpful comments, as well as for their time and effort in helping us improve the manuscript.

Reviewer #3 (Remarks to the Author):

In the revised manuscript, the authors have addressed all my previous comments. Therefore, I recommend acceptance of this manuscript without the need for further revisions.

Response: We thank the reviewer again for their helpful comments, as well as for their time and effort in helping us improve the manuscript.

Reviewer #4 (Remarks to the Author):

Thank you so much for sending this well-structured and well responded rebuttal letter back to us. I think all my questions and concerns have been answered in a decent way. I would like to recommend and to support this remarkable manuscript to be published in Nature Communications.

Han and his team has done a superb job. They not only presented massive amount of the noteworthy data, also inspire other researchers in this and related fields. This high quality paper can be a model for research in development of the ligand-targeting nucleic acid delivery systems and potentially pre-clinical therapies. Overall, this is a well-written manuscript, great job!

Response: We thank the reviewer again for their helpful comments, as well as for their time and effort in helping us improve the manuscript.